# Gender differences in submission behavior exacerbate publication disparities in elite journals

Chaoqun Ni[1], Isabel Basson[2,3], Giovanna Badia[4], Nathalie Tufenkji[5], Cassidy R Sugimoto[3,6], Vincent Larivière[2,3,7]*

[1]The Information School, University of Wisconsin-Madison, Madison, United States; [2]École de bibliothéconomie et des sciences de l'information, Université de Montréal, Montréal, Canada; [3]Department of Science and Innovation-National Research Foundation Centre of Excellence in Scientometrics and Science, Technology and Innovation Policy, Stellenbosch University, Stellenbosch, South Africa; [4]Office of the Dean of Libraries, McGill University, Montréal, Canada; [5]Department of Chemical Engineering, McGill University, Montréal, Canada; [6]School of Public Policy, Georgia Institute of Technology, Atlanta, United States; [7]Observatoire des sciences et des technologies, Université du Québec à Montréal, Montréal, Canada

*For correspondence:
vincent.lariviere@umontreal.ca

Competing interest: The authors declare that no competing interests exist.

## eLife Assessment

This **convincing** study, which is based on a survey of researchers, finds that women are less likely than men to submit articles to elite journals. It also finds that there is no relation between gender and reported desk rejection. The study is an **important** contribution to work on gender bias in the scientific literature.

**Abstract** Women are particularly underrepresented as leading authors of papers in journals of the highest impact factor, with substantial consequences for their careers. While a large body of research has focused on the outcome and the process of peer review, fewer articles have explicitly focused on gendered submission behavior and the explanations for these differences. In our study of nearly 5000 active authors, we find that women are less likely to report having submitted papers to journals of the highest impact (e.g., *Science*, *Nature*, or *PNAS*) and to submit fewer manuscripts, on average, than men when they do submit. Women were more likely to indicate that they did not submit their papers (in general and their subsequently most cited papers) to high-impact journals because they were advised not to. In the aggregate, no statistically significant difference was observed between men and women in how they rated the quality of their work. Nevertheless, regardless of discipline, women were more likely than men to indicate that their '*work was not ground-breaking or sufficiently novel*' as a rationale for not submitting to one of the listed prestigious journals. Men were more likely than women to indicate that the '*work would fit better in a more specialized journal*'. We discuss the implications of these findings and interventions that can serve to mitigate the disparities caused by gendered differences in submission behavior.

## Introduction

The rise of the research evaluation system has created a market of intense competition for a few hallowed venues. Scientific capital is strongly concentrated in generalist journals founded in the

late 19th and early 20th centuries (*Baldwin, 2015*), such as *Science* (1880), *Nature* (1869), and *Proceedings of the National Academy of Sciences of the United States of America* (PNAS, 1914). Articles published in these journals garner greater news media attention, and authors tend to receive more career opportunities and grants (*Samuel Reich, 2013*). Since publishing in highly influential journals leads to wider dissemination and visibility of researchers' work, the volume of manuscript submissions to these journals is high and acceptance low, with a large proportion of manuscripts rejected by editors without undergoing peer review. It is noted on *Science's* author portal that less than 6% of originally submitted papers are accepted (*Science, 2025*). *Nature* also notes low acceptance rates on its website, from 8% to 12% depending on the year, and states that '*many submissions are declined without being sent for review*' (*Nature, 2025*). A similar practice is espoused by the *Proceedings of the National Academy of Sciences of the United States of America* (*PNAS*), with most publications rejected before peer review and a final acceptance rate of 14.7% (*PNAS, 2025*). Prestige and hierarchy among journals are not inherently negative for scholarly communication; however, if entry into these highly visible venues varies for different sociodemographic populations of authors, it may have adverse consequences for science (*Kozlowski et al., 2022*).

Editors and peer reviewers evaluate both the quality of manuscripts submitted as well as whether the topic fits within the scope of the journal. Idealized gatekeeping reflects the Mertonian norms of organized skepticism and universalism, evaluating scientific work independently of the social characteristics of authors (*Merton, 1968*). However, various critiques of scholarly peer review have been discussed (*Ware, 2008*), among which is a concern of social biases, that is, 'differential evaluation of an author's submission as a result of her/his perceived membership in a particular social category' (*Lee et al., 2013*). These biases are not necessarily conscious; yet, if no biases were present in the case of peer review, then 'we should expect the rate with which members of less powerful social groups enjoy successful peer review outcomes to be proportionate to their representation in *submission rates*' (*Lee et al., 2013*). Results on gender disparities and bias in scholarly publishing have arrived at mixed results (*Borsuk et al., 2009*; *Day et al., 2020*; *Djupe et al., 2019*; *Edwards et al., 2018*; *Fox and Paine, 2019*; *Gilbert et al., 1994*; *Grossman, 2020*; *Helmer et al., 2017*; *Murray et al., 2018*; *Squazzoni et al., 2021*; *Walker et al., 2015*). However, many of these fail to disentangle components of the publication process, focusing only on the outcome of peer-reviewed publications rather than the effect of different groups having different submission rates or desk rejections. These components have fundamentally different attributes—with one being an effect of self-selection and the other a potential indicator of bias. We emphasize here that observations of disparities are a necessary, but insufficient indicator of bias as there are several other potential explanations for disparities. We focus, therefore, on the concept of disparity.

Most studies focused on disparities in science publishing have demonstrated that women submit fewer papers than men, with little difference in desk rejections (or a disparity in favor of women) (*Bendels et al., 2018*; *Garand and Harman, 2021*; *Grossman, 2020*; *Martinsen et al., 2022*; *Squazzoni et al., 2021*). The lower submission rate is particularly pronounced in the most prestigious journals (*Bendels et al., 2018*). Interviews with women authors explained these lower submission rates with higher selectivity: they argued that they only submit when they are confident the paper would be accepted, not wanting to risk a rejection as they have limited time for conducting research and attending to lengthy reviews (*Closa et al., 2020*). Women were also less likely than men to indicate that they submit their manuscripts to top journals in their field as their first choice and more likely than men to select a journal based on whether a journal is most likely to accept the manuscript (*Djupe et al., 2019*). Other perceptions and behavioral differences have also been postulated to account for the differences in manuscript submission behaviors, including risk aversion, perfectionism (*Borghans et al., 2009*; *Closa et al., 2020*; *Djupe et al., 2019*), and assessment of contribution (*Lincoln et al., 2012*; *Wennerås and Wold, 2008*). However, few large-scale studies have examined this question from the perspective of the authors. In this context, our paper seeks to address whether the sociodemographic characteristics of lead authors influence submission rates to elite journals. More specifically, based on a survey of more than 4700 active authors (29.9% women), we assess differences in gender, academic rank, and discipline in the likelihood of submitting to *Science*, *Nature*, and *PNAS* and the self-reported reasons for not submitting to these journals.

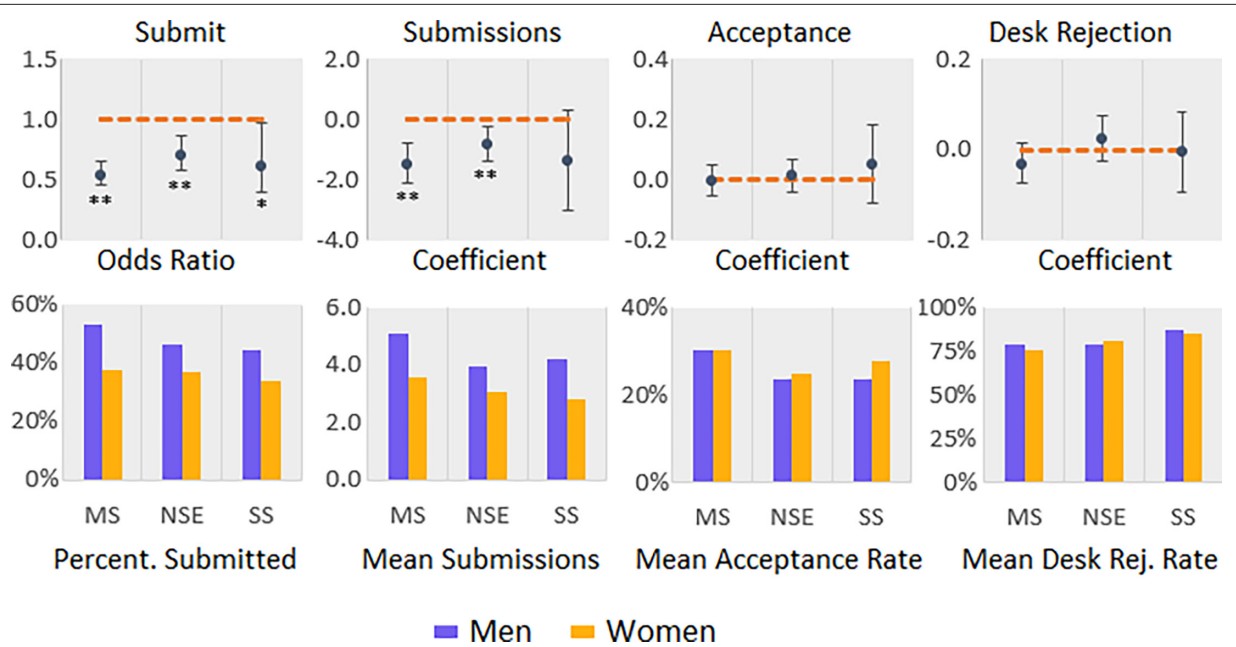

**Figure 1.** Measures of submission rate, acceptance rate, and rejection rate with tests of statistical significance. Logistic regression was used for whether they submitted to an elite journal by individual discipline. Linear regression (mixed-effect model) was used to understand the gender difference in number of submissions, acceptance rate, and desk rejection rate to these journals by individual discipline. We controlled for rank in all regression analysis. Error bars represent the 95% confidence interval. All odds ratio and coefficient values are based on women over men. MS: Medical Sciences; NSE: Natural Sciences and Engineering; SS: Social Sciences. ** indicates p < 0.01, * indicates p < 0.05.

## Results

### Journal submission behavior

Women were less likely than men to report that they submitted a paper to *Science*, *Nature*, or *PNAS* (48.7% of men and 37.0% of women, see *Appendix 1—table 7*). These differences are statistically significant (see *Appendix 1—table 8*), with men, in general, being more likely to submit to *Science*, *Nature*, or *PNAS* than women while controlling for academic rank (OR = 0.61, 95% CI [0.54, 0.70]). This relationship remains significant for each discipline (see *Figure 1*), with relatively higher percentages of men in each of the disciplines submitting to these journals (ranging from 44.2% to 53.4%), compared to women (ranging from 33.5% to 37.5%). We also investigated the mean number of manuscripts that respondents reported submitting to *Science*, *Nature*, or *PNAS* (see *Figure 1* and *Appendix 1—table 9*). All disciplines combined, 552 women and 1583 men submitted to these three elite journals. The mean was calculated by investigating only respondents who submitted at least one manuscript to *Science*, *Nature*, or *PNAS*. The number of submissions an author could have submitted depends on the discipline (due to the difference in publication behavior between disciplines) and career status (junior, senior, non-academic). In medical sciences (coefficient = −1.46, 95% CI [−2.14, −0.79], p = 0.00) and natural sciences and engineering (coefficient = −0.80, 95% CI [−1.39, −0.22], p = 0.01) women submitted fewer manuscripts than men. No statistically significant difference was observed for the social sciences (coefficient = −1.35, 95% CI [−3.00, 0.31], p = 0.11, see *Appendix 1—table 10*).

### Self-reported acceptance and rejection rate

We examined the difference between women and men regarding the self-reported acceptance and rejection rates for *Science*, *Nature*, and *PNAS*, using linear regression analysis while controlling for academic rank. The result shows no significant difference between women and men in getting their papers accepted (see *Figure 1*, *Appendix 1—table 11*, and *Appendix 1—table 12*). We also used a mixed-effect model to analyze the role of gender in the chances of getting manuscripts accepted, which yields similar conclusions to those above. We defined the desk rejection rate as the number of manuscripts that did not go out for peer review divided by the number of manuscripts submitted

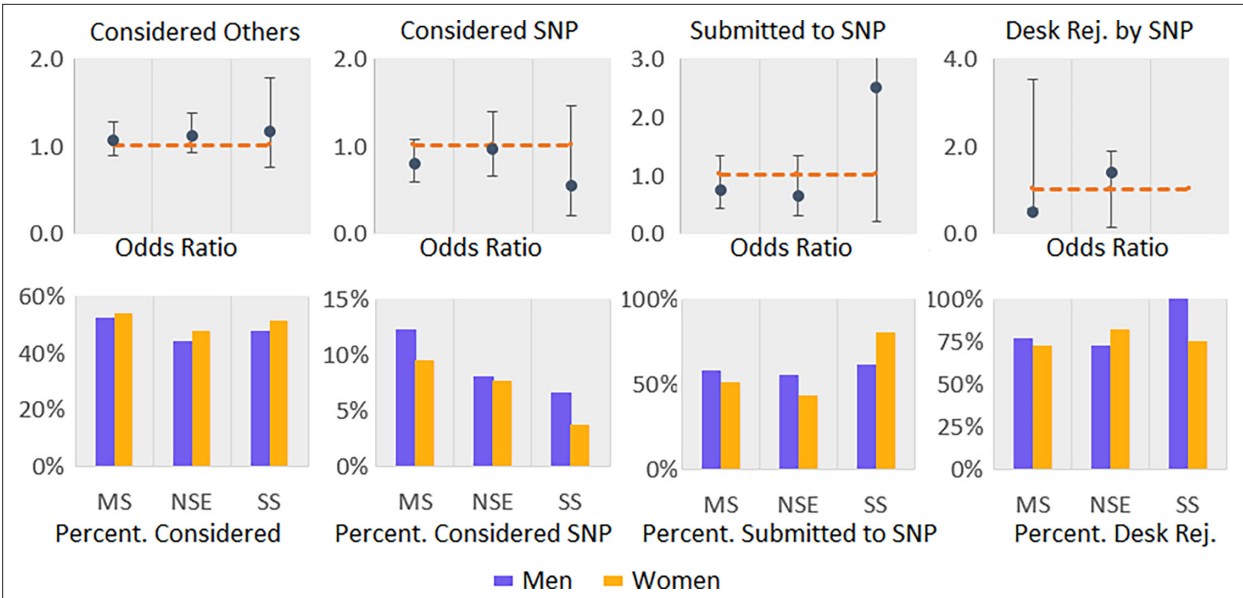

**Figure 2.** Measures of journal consideration, submission rate, and desk rejection rate for most cited papers with odds ratios (women/men). Error bars represent the 95% confidence interval. We used logistic regression to analyze the relationship between gender and the four variables, while controlling for rank. Regression was done by each discipline separately. MS: Medical Sciences; NSE: Natural Sciences and Engineering; SS: Social Sciences; SNP: Science, Nature, PNAS.

for each survey respondent. The relationship between desk rejection rate and gender was examined using linear regression and a mixed-effect model, again controlling for rank. Both models yield similar results: gender did not significantly impact the possibility of getting a desk rejection for manuscripts submitted to these journals, as illustrated in *Figure 1* (as well as *Appendix 1—table 13* and *Appendix 1—table 14*).

## Submission and desk rejections of authors' highest cited papers

To understand whether the gender differences observed above also hold for the subset of higher-impact research, we investigated respondents' submission pattern for their most cited papers (not published in *Science*, *Nature*, *PNAS*, but also *Cell*, *Nature Communications*, *NEJM*, and *Science Advances*; see Materials and methods for more details). Results show that a substantial proportion of the authors (48.6%) considered other journals than the one in which their paper was published, with no significant gender difference (see *Appendix 1—table 17* and *Appendix 1—table 18*). However, only a small proportion of authors (9.1%) considered *Science*, *Nature*, or *PNAS*, again with no significant gender difference (see *Appendix 1—table 19* and *Appendix 1—table 20*). As only a few authors considered publishing in these journals, it is not surprising to see that few authors submitted their most cited paper to *Science*, *Nature*, or *PNAS* (228 respondents), again with no significant gender gap (see *Appendix 1—table 21* and *Appendix 1—table 22*). Similarly, no significant difference was observed in desk rejection rate for their most cited paper between men and women in any of the disciplines when submitting to *Science*, *Nature*, and *PNAS* (see *Appendix 1—table 23* and *Appendix 1—table 24*). Submission behavior and the chance of getting a desk rejection for higher quality papers thus do not seem to differ significantly between men and women (*Figure 2*), although these conclusions are based on a very limited number of responses: 480 women and 1404 men across all disciplines.

## Perception of the quality of research

Respondents were asked to rate their perception of the quality of their research (not their papers) in comparison to their peers on a five-point Likert scale ranging from Excellent (5) to Poor (1). Most respondents rated their work as good or excellent (86.7%), regardless of discipline, with very few rating their work as poor or fair (1.0%), and only a few respondents rated it as average (12.4%) (*Figure 3* and *Appendix 1—table 27*). When considering responses regardless of discipline, no statistically significant difference was observed between men and women regarding how they rated the quality of their

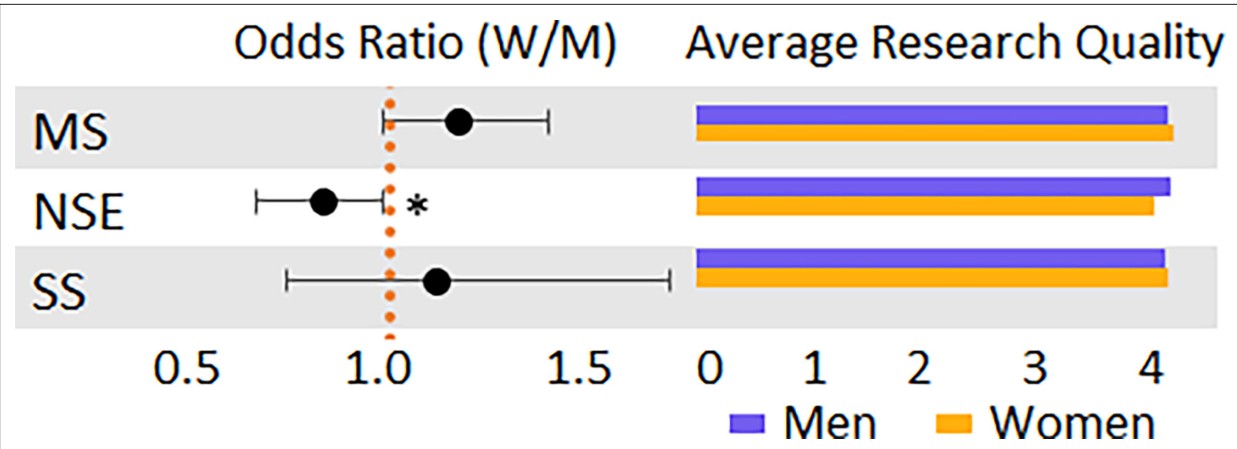

**Figure 3.** Odds ratio (women/men) and mean value for research quality comparison with peers. Ordinal logistic regression was used to analyze the relationship between gender and the rating of research quality while controlling for respondents' rank. Regression was done at the discipline level. Error bars represent the 95% confidence interval. MS: Medical Sciences; NSE: Natural Sciences and Engineering; SS: Social Sciences. * indicates p < 0.05.

work (see *Appendix 1—table 28*). When disaggregating the respondents by discipline, it was found that compared with men, women in the natural sciences and engineering are more likely to rank their research quality lower (OR = 0.83, 95% CI [0.67, 0.99], p = 0.04). No statistically significant difference was observed in the medical (OR = 1.18, 95% CI [0.99, 1.40], p = 0.07) and social sciences (OR = 1.13, 95% CI [0.93, 1.22], p = 0.58).

## Reasons for not submitting to elite journals

We also investigated the authors' rationale for why they did not submit their manuscripts (in general and their most cited) to the selected journals by asking respondents to select from a list of potential reasons. The results of the regression analysis are represented in *Figure 4*. None of the respondents with a known academic rank indicated that they did not submit to the journals because they were

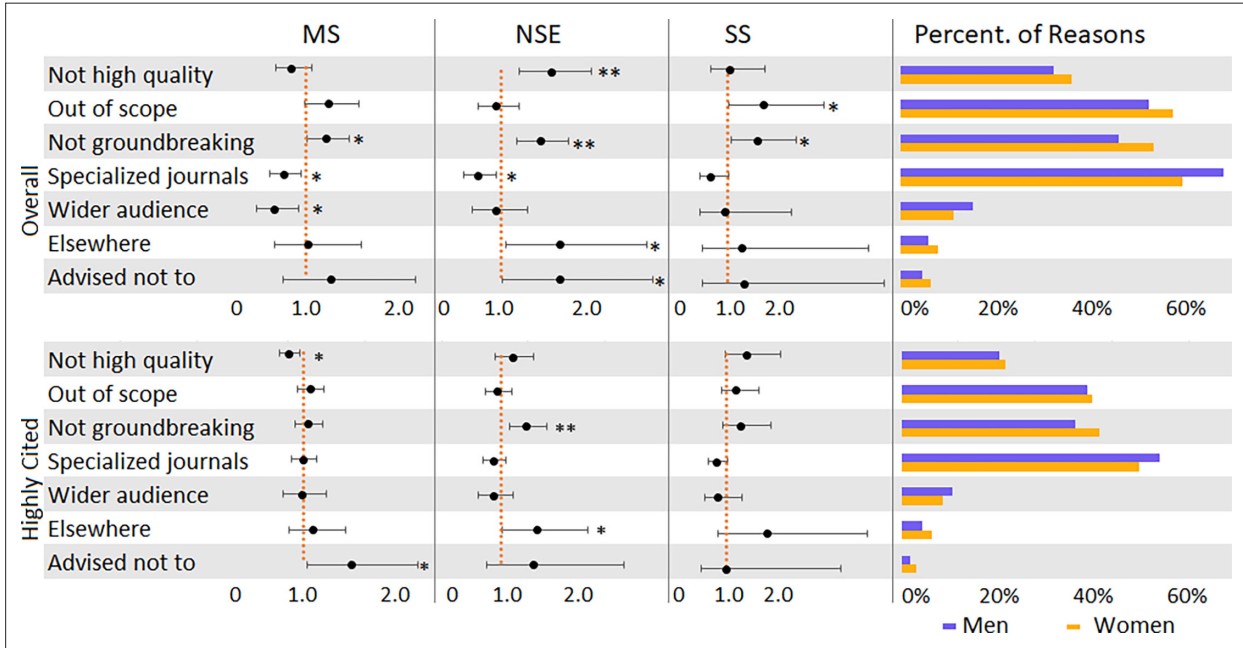

**Figure 4.** Reasons for not submitting to top journals, overall and most cited paper, with odds ratios (women/men). Logistic regression was employed to analyze the relationship between gender and each of the reasons for not submitting. Regression was done by reason and discipline while controlling for rank. Error bars represent the 95% confidence interval. MS: Medical Sciences; NSE: Natural Sciences and Engineering; SS: Social Sciences. ** indicates p < 0.01, * indicates p < 0.05.

unaware of the journals. The most common reason men and women gave was that their work would fit better in a more specialized journal.

Two of the listed reasons are related to the quality and the novelty of the research. Women in natural sciences and engineering were more likely than men to indicate that '*the work was not of high enough quality*' as one of the reasons why they did not submit to *Science*, *Nature*, or *PNAS* (OR = 1.59, 95% CI [1.22, 2.08], p = 0.001). A similar finding was also found for the most cited paper, although only significant in the medical sciences (OR = 0.75, 95% CI [0.60, 0.94], p = 0.011). Regardless of discipline, women were more likely than men to (OR = 1.26, 95% CI [1.07, 1.50]) indicate that they consider their '*work was not ground-breaking or sufficiently novel*' for the journal in question. For their most cited papers, it was also the case for authors publishing in natural sciences and engineering (OR = 1.36, 95% CI [1.10, 1.67], p = 0.004), as well as all disciplines combined (OR = 1.19, 95% CI [1.04, 1.37], p = 0.011) (see *Appendix 1—table 16* and *Appendix 1—table 26*).

Reasons related to the scope and audience of the journals mostly show non-significant gender differences, with one exception. In general, men were more likely than women to indicate that the '*work would fit better in a more specialized journal*' for their general submissions (OR = 0.72, 95% CI [0.60, 0.85], p = 0.000). When disaggregated by discipline, a significant difference was observed both for the medical sciences (OR = 0.74, 95% CI [0.57, 0.95], p = 0.018) and for the natural sciences and engineering (OR = 0.72, 95% CI [0.55, 0.95], p = 0.018). However, no statistically significant difference was observed for the most cited papers. Men in the medical sciences were also more likely than women to indicate that the '*work would reach a wider audience in another journal*' (OR = 0.61, 95% CI [0.41, 0.92], p = 0.017). No statistically significant difference was observed for the most cited papers. In contrast to the above results, women submitting in the social sciences were more likely to indicate that the '*work fell out of the scope of the journal*' (OR = 1.84, 95% CI [1.03, 3.28], p = 0.04).

The last two reasons are much less frequent and relate to the wishes of the co-authors and the advice received. In the natural sciences and engineering, women were more likely than men to indicate that their co-authors wished to submit elsewhere (OR = 1.70, 95% CI [1.06, 2.74], p = 0.029). This was also the case for their most cited paper (OR = 1.57, 95% CI [1.02, 2.40], p = 0.039). Without disaggregating by discipline, women were more likely to indicate that they did not submit their papers (in general and their most cited papers) to *Science*, *Nature*, or *PNAS* because they were advised not to. When disaggregating by discipline, this difference was also observed for respondents in natural sciences and engineering for their papers in general (OR = 1.69, 95% CI [1.02, 2.81], p = 0.042) and for medical sciences respondents for their most cited papers (OR = 1.75, 95% CI [1.05, 2.93], p = 0.032).

## Discussion

Women are particularly underrepresented in journals with the highest impact factor (*Huang et al., 2020*), with substantial consequences for their careers. While a large body of research has focused on the outcome and the process of peer review, fewer articles have explicitly focused on gendered submission behavior and the explanations for these differences. In our study of nearly 5000 active authors, we find that women are less likely to report having submitted papers and, when they have, to submit fewer manuscripts, on average, than men. This reinforces much of the literature demonstrating lower submission rates among women (*Bendels et al., 2018*; *Grossman, 2020*; *Martinsen et al., 2022*; *Squazzoni et al., 2021*). One strategy might be to optimize around homophily: women editors tend to disproportionately select women reviewers (*Helmer et al., 2017*), and women reviewers are more favorable to women-authored work (*Murray et al., 2018*). The positive reinforcement in the process will increase women reviewers and authors, which may influence submission behavior on the broader network. Another strategy might be to introduce quotas into peer review; however, this has been shown to have negative effects on women (*Leibbrandt et al., 2018*).

There is a demonstrated relationship between self-efficacy and publication output (*Hemmings and Kay, 2009*); however, our results on this were mixed. In the aggregate, no statistically significant difference was observed between men and women in how they rated the quality of their work. A difference was only observed in the natural sciences and engineering, where women ranked their research quality lower. The latter is particularly interesting, given that women tend to outperform in Engineering on several indicators (*Ghiasi et al., 2015*). The weak signal observed suggests that no particular intervention may be necessary to improve self-efficacy for this population; however, there may be some internalization of value premised on past experiences in peer review. Therefore,

addressing larger issues of bias in evaluation (*Lee et al., 2013*) may serve to mitigate any gendered differences in self-efficacy.

Regardless of discipline, women were more likely than men to indicate that their '*work was not ground-breaking or sufficiently novel*' for the listed prestigious journals. Men were more likely than women to indicate that the '*work would fit better in a more specialized journal*'. Women were more likely to indicate that they did not submit their papers (in general and their most cited papers) to *Science*, *Nature*, or *PNAS* because they were advised not to. These results may reinforce the notion of risk aversion and perfectionism (*Closa et al., 2020*), suggesting that women have a higher internal standard for what constitutes novel research. However, the more alarming result is that women seem to be deterred from submitting from people within their scholarly network. One interpretation could be that the work is truly not sufficiently novel; this would suggest that men are innately predisposed to higher novelty research designs or that they are trained in this manner. Given no evidence of the former and sufficient evidence of differential mentoring for women (*Nolan et al., 2008*), it would stand to reason that greater attention should be made to equity in research training. However, the alternative interpretation is that women are not receiving the simple encouragement necessary to seek out higher-impact venues. Research administrators may seek simple interventions to increase this support. Journal editors may also want to consider dedicated outreach to women-led labs.

Addressing gender disparities in science requires a multifaceted approach that considers all components of the scientific system. In that context, we cannot overemphasize the need for a research evaluation reform—including initiatives that span from the Leiden Manifesto to the San Francisco Declaration on Research Assessment. But until those reforms are actually adopted by universities and funders and actions move beyond pledges, publishing remains a cornerstone of this system and central to research evaluation. Therefore, addressing mechanisms that create disparities in journal submission—be they self-imposed or otherwise—is essential for creating a robust and responsible research ecosystem.

## Materials and methods

A survey was conducted to explore the relationship between author gender and manuscript submission, rejection, and acceptance rates for high-impact, multidisciplinary journals in physical sciences, life sciences, and engineering (see S.I. appendix for the questionnaire). This study focuses on the results for *Science*, *Nature*, and *PNAS*. The target population consisted of active authors who published articles from 2008 to 2017 in journals indexed by WoS in medical sciences, natural sciences and engineering, and social sciences (*Appendix 1—table 2*). The questionnaire asked respondents to report on their manuscript submission experiences. It included a subsection focusing on their most cited paper published after 2010 not published in selected elite journals, that is, *Science*, *Nature*, *PNAS* (which are analyzed in the main manuscript), as well as *Cell*, *Nature Communications*, *NEJM*, and *Science Advances* (which are analyzed in the S.I. appendix). Specific questions addressed their experiences submitting their most cited manuscripts to these journals, as well as their experiences submitting to these journals in general. Stata software (Standard Edition 17) was used to conduct the analyses, which consisted of applying ordinal logistic regression, multinomial logistic regression, and multiple logistic regression with a significance level or alpha of 0.05. Although we collected data on genders other than men and women, the independent variable studied is a binary gender variable (i.e., women or men), given the low number of respondents outside these two categories. Respondents were disaggregated by academic rank (i.e., junior, senior, or non-academic, excluding all unknowns) and discipline (i.e., social sciences, medical sciences, and natural sciences and engineering). More details about the regression analysis can be found in the appendix. The sample consisted of 6002 respondents who participated in the survey, of whom 4857 finished the questionnaire. An analysis of the attrition failed to identify a common point of departure, suggesting individual variability in dropout rather than failed survey construction. The final number of respondents in this study is 4805, after the removal of 19 responses in Arts and Humanities and additional responses due to a lack of information for critical variables. Of the final number of respondents, 4740 (98.6%) have a known academic rank and are included in the regression analysis (more details available in the S.I. *Appendix 1—table 6*).

## Acknowledgements

Vincent Larivière acknowledges funding from the Canada Research Chairs program. The authors would like to thank the survey respondents and the Indiana University Center for Survey Research for their critical contribution to the study. This analysis was approved by McGill University's Research Ethics Board II, R.E.B. File #: 501-0518 *Factors affecting scientific manuscript submission and acceptance rates.*

## Additional information

### Funding

| Funder | Grant reference number | Author |
|---|---|---|
| Canada Research Chairs | | Vincent Larivière |

The funders had no role in study design, data collection, and interpretation, or the decision to submit the work for publication.

### Author contributions

Chaoqun Ni, Data curation, Software, Formal analysis, Validation, Investigation, Visualization, Writing – review and editing; Isabel Basson, Investigation, Writing – original draft, Writing – review and editing; Giovanna Badia, Nathalie Tufenkji, Conceptualization, Methodology, Writing – review and editing; Cassidy R Sugimoto, Conceptualization, Methodology, Writing – original draft, Writing – review and editing; Vincent Larivière, Conceptualization, Resources, Supervision, Funding acquisition, Investigation, Methodology, Writing – original draft, Project administration, Writing – review and editing

### Author ORCIDs

Chaoqun Ni ⓘ https://orcid.org/0000-0002-4130-7602
Isabel Basson ⓘ https://orcid.org/0000-0001-6150-3208
Giovanna Badia ⓘ https://orcid.org/0000-0003-1544-2728
Nathalie Tufenkji ⓘ https://orcid.org/0000-0002-1546-3441
Vincent Larivière ⓘ https://orcid.org/0000-0002-2733-0689

### Ethics

This analysis was approved by McGill University's Research Ethics Board II, R.E.B. File #: 501-0518 Factors affecting scientific manuscript submission and acceptance rates.

Joint Public Review: https://doi.org/10.7554/eLife.90049.4.sa1
Author response https://doi.org/10.7554/eLife.90049.4.sa2

## Additional files

### Supplementary files

MDAR checklist

### Data availability

All data needed to evaluate the conclusions in the paper are present in the paper and the Supporting information. Aggregated, de-identified data by gender, discipline, and rank for analyses are available on GitHub (https://github.com/MetascienceLab/genderPublishing/ copy archived at *Metascience Research Lab, 2024*). Due to Institutional Review Board restrictions and the terms of participant consent, the survey data used in this study cannot be shared publicly. However, to support transparency and reproducibility, all analysis code and a synthetic dataset- constructed with randomly generated values but matching the original data structure- are available on Zenodo: https://doi.org/10.5281/zenodo.16327580.

The following dataset was generated:

| Author(s) | Year | Dataset title | Dataset URL | Database and Identifier |
|---|---|---|---|---|
| Ni C, Basson I, Badia G, Tufenkji N, Sugimoto CR, Larivière V | 2025 | Data for Gender differences in submission behavior exacerbate publication disparities in elite journals | https://doi.org/10.5281/zenodo.16327580 | Zenodo, 10.5281/zenodo.16327580 |

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

# Appendix 1

## Expansion on materials and methods

### Sampling frame construction

The sampling frame was constructed using data extracted from Web of Science (WoS), as hosted by the Centre for Science and Technology Studies (CWTS) at Leiden University. Using the author disambiguation algorithm developed by Caron and van Eck (2014) (Caron, E. & Van Eck, N.J. (2014). "Large scale author name disambiguation using rule-based scoring and clustering." in Proceedings of the 19th international conference on science and technology indicators, Leiden), we extracted metadata of all publications for authors who have been corresponding authors of at least one paper between 2008 and 2017. The metadata included the email addresses of these authors; these email addresses were used to contact the participant population. A minimum cut-off point of 10 papers for the identification of prolific authors was selected. When considering whether an author is a prolific author, only research papers were considered. The total study population consists of 494,781 prolific authors, data on the gender and discipline in which they publish were known for 415,697 authors (84.0%). The gender of an author for the sampling frame was determined by the algorithm developed in Larivière et al. (2013) (Larivière, V., Ni, C., Gingras, Y., Cronin, B. & Sugimoto, C.R. (2013). "Bibliometrics: Global gender disparities in science." Nature, 504: 211–213. doi:10.1038/504211a) whereas the gender variable used in the analysis was determined by the responses of the authors. The discipline in which an author publishes was determined by the WoS journal classification.

### Identifying most cited papers

Several questions in the questionnaire ask the authors about their decisions pertaining to their most cited paper. The questionnaires are customized to each respondent, showing them the title of their most cited paper when these questions are asked. The most cited paper was identified from the previous corpus of papers by the author published between 2008 and 2017, that has received the most citations at the time of data extraction and were not published in the selected elite journals. Despite the fact that citations take time to accumulate, the skewed nature of citation distributions—where most papers remain lowly cited, and a few papers get highly cited—leads to a situation where most cited papers are not necessarily the oldest papers published by respondents.

### Elite journal selection

The questionnaire mainly consists of questions related to the author's submission behavior as they pertain to elite journals. The seven journals were chosen given their general reputation and research impact. All seven journals were in the top five journals with the highest impact factors in the 2016 Journal Citation Reports categories of 'Multidisciplinary sciences', 'General & internal medicine', and 'Biology'. Journal Citation Reports (more information at https://clarivate.libguides.com/jcr) (https://jcr.clarivate.com/jcr/) is a resource published annually by Clarivate Analytics that provides the impact factor for all journals included in WoS. The journal impact factor (Garfield, E. (2006). "The history and meaning of the journal impact factor." JAMA, 295(1):90–93. doi:10.1001/jama.295.1.90) is a calculation that employs citations that articles in a journal received as an indication of the influence of that journal in its field.

### Regression analysis

This study used regression analysis to analyze the data. The analysis described here is similar to **Ni et al., 2021** and how it is reported in its supplementary materials. Here, the discussion is slightly adapted to reflect the variables and questions used within this study. The regression procedures include binary logistic regression, ordinal logistic regression, linear regression, and mixed-effects model. Regression analysis is usually used to explore relationships between dependent and independent variables. The most common is linear regression, in which:

$$Y_i = \beta_0 + \beta_1 X_i + \beta_2 Z_i + \cdots + e_i,$$

where with one unit change in $X$ there is $\beta_1$ differences in $Y$ after $Z$ and other variables are controlled. However, this approach assumes that all dependent variables are normally distributed. Furthermore, it often assumes that the dependent variable $Y$ is continuous. Our dataset, like many social science surveys, is replete with categorical variables. We want to understand whether there is a gendered difference when selecting one category over another. Therefore, we turned to logistic regression. Specific procedures and analysis methods vary by the scale of dependent variables, as well as the number of variable categories.

For questions with answers measured in ordinal scales (e.g., Likert scaling), ordinal logistic regression analysis was employed. For questions with answers (the dependent variables) presented in a categorical, unordered scale, binary logistic regression (when the dependent variable only contains two categories) was performed. The point statistic was still the odds ratio of women over men. Here again, gender is the independent variable, with the academic rank being the controlled variable in the case when a dependent variable only has two categories (e.g., 'yes' or 'no' to an answer), binary logistic regression analysis was conducted to analyze the data. An odds ratio value significantly over 1 indicates that women were more likely than men to select 'Yes' for this question. A separate regression analysis is performed for each discipline with rank being controlled. For analyses concerning all disciplines, we control for both academic rank and discipline.

The binary logistic regression model for each discipline can be described as follows:

$$Y_i = \text{Log}\left(\frac{p}{1-p}\right) = \beta_0 + \beta_1 X_i + \beta_2 Z_i + \cdots + e_i,$$

where $Y_i$ is the log probability of the binary dependent variable (such as whether you submitted to a journal), $X_i$ is the binary independent variable gender (woman = 1 and man = 0), and $Z_i$ is the other control variable(s).

The mixed-effect linear model for each discipline can be described as follows:

$$Y_i = \beta_0 + \beta_1 X_i + Z_i + \cdots + e_i,$$

where $Y_i$ is the dependent variable (number of submissions, acceptance rate, or rejection rate), $X_i$ is the gender of the individual (the fixed effect), and $Z_i$ is the rank of the individual (random effect variable).

## Recoded categories

The academic rank variable was created by consulting the respondents' answers on the questions regarding academic rank (for those respondents that indicated their primary sector affiliation as Academia), and position (for those respondents from the Government, Private, or Not_for_Profit sector). Responses were recategorized into junior, senior, and non-academic, with unknowns removed from analyses, as per *Appendix 1—table 1*.

**Appendix 1—table 1.** Recategorized academic rank.

| Questionnaire categories | New categories | Count |
|---|---|---|
| Graduate student | Junior | 991 |
| Postdoctoral fellow | | |
| Research Associate | | |
| Assistant Professor | | |
| Associate Professor | Senior | 3012 |
| Full Professor | | |
| Emeritus Professor | | |
| Non-academic respondents | Non-academic | 737 |
| No answer and other | Unknowns | 65 |

Due to limited responses, analysis per journal was not always viable. The results pertaining to the journals were aggregated, with new categories based on the shared similarities in disciplinary foci of the journals and their prestige. The categories are as follows:

- Science, Nature, and PNAS (S.N.P.)
- Nature Communications and Science Advances (NC.SA.)
- NEJM and Cell (NEJM.C.)

Similarly, the various options for research areas in the questionnaire were reclassified into three broader areas that share common research and dissemination practices, namely natural science and engineering (NSE), medical science (MS), and social sciences (SS), as shown in *Appendix 1—table 2*.

**Appendix 1—table 2.** Recategorized research areas to disciplines.

| Disciplinary area | Field | Surveyed | Respondents | Analytical sample |
|---|---|---|---|---|
| Medical Sciences | Biomedical Research | 63,335 | 832 | 678 |
| | Clinical Medicine | 152,056 | 1394 | 1134 |
| | Health | 12,807 | 245 | 208 |
| Natural Sciences and Engineering | Biology | 47,539 | 670 | 543 |
| | Chemistry | 36,830 | 395 | 321 |
| | Earth and Space | 36,573 | 576 | 467 |
| | Engineering and Technology | 61,746 | 559 | 406 |
| | Mathematics | 18,674 | 275 | 206 |
| | Physics | 46,694 | 592 | 475 |
| Social Sciences | Professional Fields | 3480 | 82 | 66 |
| | Psychology | 9885 | 243 | 207 |
| | Social Sciences | 4354 | 114 | 94 |
| Arts and Humanities | Arts | 291 | 9 | / |
| | Humanities | 475 | 14 | / |
| Unknown | Unknown | 42 | 2 | 0 |
| | Total | 494,781 | 6002 | 4805 |

## Description of the analytical sample

The Indiana University Centre for Survey Research (CSR) was contracted to administer the survey. The description of the survey response is based on the technical report produced by the CSR. After data cleaning (e.g., removing duplicates), a total of 489,777 email invitations were sent, and 6002 individuals participated in the online survey. The responses consisted of 4857 completed questionnaires. After reviewing the complete questionnaires, responses of authors in Arts (8) and humanities (11) questionnaires were excluded. Lastly, we further removed additional responses due to lack of information for critical variables (e.g., gender and discipline). An analysis of the attrition failed to identify a common point of departure, suggesting individual variability in dropout rather than failed survey construction. The final analytical sample consists of 4805 respondents. Depending on the variable, the number of responses for analyzing them may vary.

The response rate of the survey was 1.2%, as classified according to The American Association for Public Opinion Research, 2015. Standard Definitions: Final Dispositions of Case Codes and Outcome Rates for Surveys. 8th edition (AAPOR), as summarized in *Appendix 1—table 3*.

**Appendix 1—table 3.** Summary of response rate of questionnaire adapted from McGill University – Gender Ethnicity Publishing Study Methods Summary technical report (Indiana University Centre for Survey Research 2019).
McGill University – Gender Ethnicity Publishing Study Methods Summary. Technical Report. Unpublished.

| Disposition | Definition | Count | Response rate (RR2) |
|---|---|---|---|
| Complete (I) | Respondent completed survey | 4944 | |
| Partial (P) | Respondent answered at least one question item but did not complete survey | 1058 | |
| Implicit Refusal (R) | Respondent clicked survey link but did not answer any items | 1655 | |
| Nothing Returned (UH) | Respondent did not respond to survey; it is unknown if any email messages were read | 374,341 | 1.2% |
| Undeliverable (UO) | Recruitment message was not received by intended recipient due to email and/ or mailing returns | 108,163 | |
| Not Eligible | Sample member indicated they were not eligible to participate; also includes cases of authors who have deceased since their publication listed in the database | 58 | |
| Total | | 490,219 | |

*Appendix 1—tables 4–6* provide a description of the population, respondents, and analytical sample by gender.

**Appendix 1—table 4.** Gender composition of population, respondents, and analysis sample.

| Domain | Population/surveyed | | Respondents | | Analytical sample | |
|---|---|---|---|---|---|---|
| | N | % | N | % | N | % |
| Men | 286,571 | 57.9 | 3508 | 58.4 | 2832 | 58.9 |
| Women | 129,592 | 26.2 | 1774 | 29.6 | 1435 | 29.9 |
| Unknown | 78,618 | 15.9 | 720 | 12.0 | 538 | 11.2 |
| All | 494,781 | 100.0 | 6002 | 100.0 | 4805 | 100.0 |

**Appendix 1—table 5.** Discipline composition of population, respondents, and analysis sample.

| Domain | Population/surveyed | | Respondents | | Analytical sample | |
|---|---|---|---|---|---|---|
| | N | % | N | % | N | % |
| MS | 228,198 | 46.1 | 2471 | 41.2 | 2020 | 42.0 |
| SS | 17,719 | 3.6 | 439 | 7.3 | 366 | 7.6 |
| NSE | 248,098 | 50.1 | 3069 | 51.1 | 2419 | 50.3 |
| AH | 766 | 0.2 | 23 | 0.4 | | 0.0 |
| All disciplines | 494,781 | 100.0 | 6002 | 100.0 | 4805 | 100.0 |

**Appendix 1—table 6.** Discipline, gender, and academic rank composition of analytical sample.

| Rank/role | MS (N = 2020) | | | | NSE (N = 2419) | | | | SS (N = 366) | | | |
|---|---|---|---|---|---|---|---|---|---|---|---|---|
| | Women | | Men | | Women | | Men | | Women | | Men | |
| | N | % | N | % | N | % | N | % | N | % | N | % |
| Junior | 200 | 24.5 | 228 | 18.9 | 147 | 27.6 | 360 | 19.1 | 31 | 18.7 | 25 | 12.5 |

*Appendix 1—table 6 Continued on next page*

*Appendix 1—table 6 Continued*

|  | MS (*N* = 2020) | | | | NSE (*N* = 2419) | | | | SS (*N* = 366) | | | |
|---|---|---|---|---|---|---|---|---|---|---|---|---|
| Senior | 465 | 57.0 | 762 | 63.3 | 281 | 52.7 | 1219 | 64.6 | 124 | 74.7 | 161 | 80.5 |
| Non-academic | 131 | 16.1 | 189 | 15.7 | 100 | 18.8 | 294 | 15.6 | 10 | 6.0 | 13 | 6.5 |
| Unknown | 20 | 2.5 | 25 | 2.1 | 5 | 0.9 | 13 | 0.7 | 1 | 0.6 | 1 | 0.5 |

## Limitations

Several limitations should be noted regarding the findings of this study. First, the survey response rate is less than 2%, which may limit the generalizability of the findings. In addition, asking researchers about their behavior and intentions regarding their manuscript submissions retroactively can be complicated. There is a concern related to the capacity of researchers to remember their motivations at the time of submission; researchers with longer careers—and therefore a higher number of papers—may have submitted to various journals for various reasons. This would also make it difficult to ascertain the reasons why they do not submit to the journals, as the motivations would be different for each paper. Moreover, reasons for not submitting to these journals were not exhaustive, as noted by the various comments made by respondents. One of the notable reasons given by respondents that relate to the gender component of the study is that they did not submit to these journals due to the low percentage of women on the journal editorial boards. Based on respondents' comments, page fees and article processing charges (APC) were also a serious consideration for some authors when deciding where to publish their work. Unfortunately, this consideration was not incorporated into the listed reasons in the questionnaire.

There are also several factors pertaining to the nature of most cited papers that complicate the interpretation of the results. For example, some respondents stated that their most cited paper was an invited paper of some kind, thus the choice of journal was already made for them and, therefore, the survey questions were irrelevant. In addition, even though their most cited paper was used as a proxy of a high-quality paper, the paper in question was not necessarily a highly cited paper, or necessarily significantly more cited than their other papers, or a paper of high quality (e.g., all their other papers could have been cited once and their most cited paper twice). That being said, the papers analyzed had, on average, a research impact 2.3 times higher than the world average of their field, which shows that they can be categorized as high impact. On the whole, this was the best proxy available for assignment of high-quality papers to researchers, and the effect is reduced by only investigating the subset of prolific researchers.

Some limitations relate to the passage of time between manuscript submission and data collection. Academic rank refers to the academic rank of the author at the time of the survey and not at the time of manuscript submission. This could result in a scenario where a respondent's highest cited paper could have been submitted in 2008 while they were a junior academic but in 2019, at the time of the survey, they have since changed academic rank to senior academic. Therefore, this should be interpreted as the maximum rank reached by the respondent. Review policies of journals are also a potential limitation as data pertaining to the review policies at the time of manuscript submission were not collected. If a journal had a double-blind review policy at the time of submission, it might reduce potential gender bias and be a confounding effect if journals of different peer review types were grouped together. It is likely that the review policy of the journals was single-blind at the time of submission of the papers as it is the review policy most used in the fields of life sciences, physical sciences, and engineering (Ware, M. (2008). "Peer review in scholarly journals: Perspective of the scholarly community — an international study." Information Services & Use, 28(2). doi:10.5555/1454388.1454399) and thus unlikely to have changed. While this could be a factor when we investigated acceptance rate, it is not a consideration when investigating desk rejections as none of the journals include an editor-blinded review policy.

## Regression and additional descriptive tables

The questionnaire was divided into four parts, and the following sections are ordered by the sections of the questionnaire. The sections are as follows:

1. Part 1: Consent for Research Study Participation (information regarding questionnaire – no questions)
2. Part 2: Your manuscript submissions (i.e., questions pertaining to submissions in general)
3. Part 3: Your journal article (i.e., questions pertaining to most cited paper)
4. Part 4: Demographics (as well as a question pertaining to quality of research in general and open comments)

## Part 2: Submissions in general
Journal submission behavior

**Appendix 1—table 7.** Question: 'For each of the journals listed in the table below, please indicate the appropriate number of manuscripts submitted'—percentage by discipline, journal, and gender ever submitted.

| S.N.P. | Women | | | | Men | | | |
|---|---|---|---|---|---|---|---|---|
| | No | Yes | No response | %Yes | No | Yes | No response | %Yes |
| MS | 504 | 303 | 9 | 37.5% | 555 | 636 | 13 | 53.4% |
| NSE | 328 | 194 | 11 | 37.2% | 1000 | 860 | 26 | 46.2% |
| SS | 109 | 55 | 2 | 33.5% | 110 | 87 | 3 | 44.2% |
| All disciplines | 941 | 552 | 22 | 37.0% | 1665 | 1583 | 42 | 48.7% |
| NC.SA. | Women | | | | Men | | | |
| | No | Yes | No response | %Yes | No | Yes | No response | %Yes |
| MS | 670 | 137 | 9 | 17.0% | 902 | 283 | 19 | 23.9% |
| NSE | 429 | 91 | 13 | 17.5% | 1476 | 375 | 35 | 20.3% |
| SS | 158 | 5 | 3 | 3.1% | 181 | 14 | 5 | 7.2% |
| All disciplines | 1257 | 233 | 25 | 15.6% | 2559 | 672 | 59 | 20.8% |
| NEJM.C | Women | | | | Men | | | |
| | No | Yes | No response | %Yes | No | Yes | No response | %Yes |
| MS | 603 | 204 | 9 | 25.3% | 795 | 390 | 19 | 32.9% |
| NSE | 505 | 12 | 16 | 2.3% | 1816 | 34 | 36 | 1.8% |
| SS | 152 | 11 | 3 | 6.7% | 182 | 13 | 5 | 6.7% |
| All disciplines | 1260 | 227 | 28 | 15.3% | 2793 | 437 | 60 | 13.5% |

**Appendix 1—table 8.** Question: 'For each of the journals listed in the table below, please indicate the appropriate number of manuscripts submitted'—odds ratio (women to men) values for the probability of submitting.

| Journal | Discipline | Odds ratio | Std. Err. | z | p > \|z\| | 95% CI lower | 95% CI upper | obs |
|---|---|---|---|---|---|---|---|---|
| | MS | 0.54 | 0.05 | −6.45 | 0.000 | 0.45 | 0.65 | 1937 |
| | NSE | 0.70 | 0.07 | −3.40 | 0.001 | 0.58 | 0.86 | 2356 |
| | SS | 0.62 | 0.14 | −2.13 | 0.033 | 0.40 | 0.96 | 355 |
| S.N.P. | All disciplines | 0.61 | 0.04 | −7.32 | 0.000 | 0.54 | 0.70 | 4648 |
| | MS | 0.65 | 0.08 | −3.68 | 0.000 | 0.52 | 0.82 | 1931 |
| | NSE | 0.81 | 0.11 | −1.60 | 0.110 | 0.63 | 1.05 | 2346 |
| | SS | 0.35 | 0.19 | −1.91 | 0.056 | 0.12 | 1.02 | 352 |
| NC.SA. | All disciplines | 0.70 | 0.06 | −4.06 | 0.000 | 0.59 | 0.83 | 4629 |

*Appendix 1—table 8 Continued on next page*

*Appendix 1—table 8 Continued*

| Journal | Discipline | Odds ratio | Std. Err. | z | p > |z| | 95% CI lower | 95% CI upper | obs |
|---|---|---|---|---|---|---|---|---|
| | MS | 0.71 | 0.07 | −3.35 | 0.001 | 0.58 | 0.87 | 1931 |
| | NSE | 1.22 | 0.42 | 0.59 | 0.558 | 0.63 | 2.39 | 2342 |
| | SS | 1.08 | 0.46 | 0.18 | 0.859 | 0.47 | 2.50 | 352 |
| NEJM.C. | All disciplines | 0.75 | 0.07 | −2.95 | 0.003 | 0.62 | 0.91 | 4625 |

**Appendix 1—table 9.** Question: 'For each of the journals listed in the table below, please indicate the appropriate number of manuscripts submitted'—mean number of manuscript submissions.

| Average number of submissions to S.N.P. | Women | | | | | Men | | | | |
|---|---|---|---|---|---|---|---|---|---|---|
| | Obs | Mean | Std. dev. | Min | Max | Obs | Mean | Std. dev. | Min | Max |
| MS | 303 | 3.60 | 3.51 | 1 | 34 | 636 | 5.12 | 5.41 | 1 | 35 |
| NSE | 194 | 3.04 | 2.38 | 1 | 13 | 860 | 3.95 | 3.95 | 1 | 30 |
| SS | 55 | 2.80 | 3.36 | 1 | 19 | 87 | 4.22 | 5.49 | 1 | 30 |
| All disciplines | 552 | 3.32 | 3.15 | 1 | 34 | 1583 | 4.43 | 4.71 | 1 | 35 |

| Average number of submissions to NC.SA. | Women | | | | | Men | | | | |
|---|---|---|---|---|---|---|---|---|---|---|
| | Obs | Mean | Std. dev. | Min | Max | Obs | Mean | Std. dev. | Min | Max |
| MS | 137 | 1.74 | 1.34 | 1 | 10 | 281 | 1.90 | 1.24 | 1 | 8 |
| NSE | 89 | 1.65 | 1.08 | 1 | 6 | 374 | 1.86 | 1.57 | 1 | 17 |
| SS | 5 | 1.00 | 0.00 | 1 | 1 | 14 | 1.71 | 1.14 | 1 | 5 |
| All disciplines | 231 | 1.69 | 1.23 | 1 | 10 | 14 | 1.71 | 1.14 | 1 | 5 |

| Average number of submissions to NEJM.C. | Women | | | | | Men | | | | |
|---|---|---|---|---|---|---|---|---|---|---|
| | Obs | Mean | Std. dev. | Min | Max | Obs | Mean | Std. dev. | Min | Max |
| MS | 203 | 1.89 | 1.34 | 1 | 10 | 389 | 2.25 | 2.00 | 1 | 16 |
| NSE | 12 | 1.67 | 1.44 | 1 | 6 | 34 | 1.76 | 1.48 | 1 | 8 |
| SS | 11 | 1.64 | 0.92 | 1 | 4 | 13 | 1.15 | 0.38 | 1 | 2 |
| All disciplines | 226 | 1.86 | 1.32 | 1 | 10 | 436 | 2.18 | 1.95 | 1 | 16 |

**Appendix 1—table 10.** Question: 'For each of the journals listed in the table below, please indicate the appropriate number of manuscripts submitted'—mixed linear regression.

| Journal | Discipline | Coefficient | Std. dev. | t | p > |t| | 95% CI lower | 95% CI upper | obs |
|---|---|---|---|---|---|---|---|---|
| | MS | −1.46 | 0.34 | −4.26 | 0.000 | −2.14 | −0.79 | 921 |
| | NSE | −0.80 | 0.30 | −2.69 | 0.007 | −1.39 | −0.22 | 1042 |
| | SS | −1.35 | 0.84 | −1.61 | 0.110 | −3.00 | 0.31 | 140 |
| S.N.P. | All disciplines | −1.19 | 0.22 | −5.41 | 0.000 | −1.62 | −0.76 | 2103 |
| | MS | −0.19 | 0.13 | −1.41 | 0.161 | −0.45 | 0.08 | 411 |
| | NSE | −0.19 | 0.18 | −1.09 | 0.276 | −0.54 | 0.16 | 459 |
| | SS | −0.71 | 0.54 | −1.32 | 0.207 | −1.86 | 0.44 | 19 |
| NC.SA. | All disciplines | −0.20 | 0.11 | −1.89 | 0.059 | −0.42 | 0.01 | 889 |

*Appendix 1—table 10 Continued on next page*

*Appendix 1—table 10 Continued*

| Journal | Discipline | Coefficient | Std. dev. | t | p > |t| | 95% CI lower | 95% CI upper | obs |
|---------|-----------|-------------|-----------|------|--------|--------------|--------------|-----|
| | MS | –0.36 | 0.15 | –2.39 | 0.017 | –0.65 | –0.06 | 575 |
| | NSE | –0.08 | 0.50 | –0.17 | 0.867 | –1.10 | 0.93 | 46 |
| | SS | 0.59 | 0.32 | 1.84 | 0.081 | –0.08 | 1.26 | 24 |
| NEJM.C. | All disciplines | –0.31 | 0.14 | –2.23 | 0.026 | –0.58 | –0.04 | 645 |

## Acceptance rate

**Appendix 1—table 11.** Question: 'For each of the journals listed in the table below, please indicate the appropriate number of manuscripts accepted'—mean acceptance rate.

| Acceptance rate of S.N.P. | Women | | | | | Men | | | | |
|---------------------------|------|------|-----------|-----|-----|------|------|-----------|-----|-----|
| | Obs | Mean | Std. dev. | Min | Max | Obs | Mean | Std. dev. | Min | Max |
| MS | 262 | 0.30 | 0.35 | 0 | 1 | 565 | 0.30 | 0.34 | 0 | 1 |
| NSE | 177 | 0.25 | 0.35 | 0 | 1 | 798 | 0.24 | 0.34 | 0 | 1 |
| SS | 53 | 0.28 | 0.38 | 0 | 1 | 80 | 0.24 | 0.36 | 0 | 1 |
| All disciplines | 492 | 0.28 | 0.35 | 0 | 1 | 1443 | 0.26 | 0.34 | 0 | 1 |

| Acceptance rate of NC.SA. | Women | | | | | Men | | | | |
|---------------------------|------|------|-----------|-----|-----|------|------|-----------|-----|-----|
| | Obs | Mean | Std. dev. | Min | Max | Obs | Mean | Std. dev. | Min | Max |
| MS | 137 | 0.27 | 0.39 | 0 | 1 | 281 | 0.32 | 0.39 | 0 | 1 |
| NSE | 89 | 0.30 | 0.41 | 0 | 1 | 374 | 0.36 | 0.42 | 0 | 1 |
| SS | 5 | 0.20 | 0.45 | 0 | 1 | 14 | 0.24 | 0.42 | 0 | 1 |
| All disciplines | 231 | 0.28 | 0.40 | 0 | 1 | 669 | 0.34 | 0.41 | 0 | 1 |

| Acceptance rate of NEJM.C. | Women | | | | | Men | | | | |
|----------------------------|------|------|-----------|-----|-----|------|------|-----------|-----|-----|
| | Obs | Mean | Std. dev. | Min | Max | Obs | Mean | Std. dev. | Min | Max |
| MS | 202 | 0.27 | 0.38 | 0 | 1 | 388 | 0.26 | 0.36 | 0 | 1 |
| NSE | 12 | 0.17 | 0.33 | 0 | 1 | 34 | 0.18 | 0.34 | 0 | 1 |
| SS | 11 | 0.39 | 0.47 | 0 | 1 | 13 | 0.04 | 0.14 | 0 | 0.5 |
| All disciplines | 225 | 0.27 | 0.38 | 0 | 1 | 435 | 0.25 | 0.36 | 0 | 1 |

**Appendix 1—table 12.** Question: 'For each of the journals listed in the table below, please indicate the appropriate number of manuscripts accepted'—linear regression.

| Journal | Discipline | Coefficient | Std. dev. | t | p > |t| | 95% CI lower | 95% CI upper | obs |
|---------|-----------|-------------|-----------|------|--------|--------------|--------------|-----|
| | MS | 0.00 | 0.03 | 0.01 | 0.989 | –0.05 | 0.05 | 810 |
| | NSE | 0.01 | 0.03 | 0.48 | 0.628 | –0.04 | 0.07 | 964 |
| | SS | 0.05 | 0.07 | 0.78 | 0.435 | –0.08 | 0.18 | 132 |
| S.N.P. | All disciplines | 0.01 | 0.02 | 0.57 | 0.566 | –0.03 | 0.05 | 1906 |
| | MS | –0.05 | 0.04 | –1.15 | 0.252 | –0.13 | 0.03 | 411 |
| | NSE | –0.07 | 0.05 | –1.42 | 0.156 | –0.17 | 0.03 | 459 |
| | SS | –0.04 | 0.22 | –0.16 | 0.877 | –0.51 | 0.44 | 19 |
| NC.SA. | All disciplines | 0.06 | 0.03 | –1.78 | 0.075 | –0.12 | 0.01 | 889 |

*Appendix 1—table 12 Continued on next page*

*Appendix 1—table 12 Continued*

| Journal | Discipline | Coefficient | Std. dev. | t | p > \|t\| | 95% CI lower | 95% CI upper | obs |
|---------|-----------|-------------|-----------|------|-------|--------------|--------------|-----|
|  | MS | 0.02 | 0.03 | 0.65 | 0.517 | –0.04 | 0.08 | 573 |
|  | NSE | –0.02 | 0.11 | –0.15 | 0.885 | –0.25 | 0.21 | 46 |
|  | SS | 0.43 | 0.15 | 2.79 | 0.011 | 0.11 | 0.74 | 24 |
| NEJM.C. | All disciplines | 0.03 | 0.03 | 1.07 | 0.287 | –0.03 | 0.09 | 643 |

## Desk rejections

**Appendix 1—table 13.** Question: 'For each of the journals listed in the table below, please indicate the appropriate number of manuscripts rejected'—mean desk rejection rate.

| Desk rejection rate of S.N.P. | Women | | | | | Men | | | | |
|------------------------------|-----|------|-----------|-----|-----|------|------|-----------|-----|-----|
|  | Obs | Mean | Std. dev. | Min | Max | Obs | Mean | Std. dev. | Min | Max |
| MS | 262 | 0.76 | 0.34 | 0 | 1 | 565 | 0.79 | 0.28 | 0 | 1 |
| NSE | 172 | 0.81 | 0.31 | 0 | 1 | 764 | 0.79 | 0.29 | 0 | 1 |
| SS | 46 | 0.85 | 0.26 | 0 | 1 | 75 | 0.87 | 0.24 | 0 | 1 |
| All disciplines | 480 | 0.79 | 0.32 | 0 | 1 | 1404 | 0.80 | 0.28 | 0 | 1 |

| Desk rejection rate of NC.SA. | Women | | | | | Men | | | | |
|------------------------------|-----|------|-----------|-----|-----|------|------|-----------|-----|-----|
|  | Obs | Mean | Std. dev. | Min | Max | Obs | Mean | Std. dev. | Min | Max |
| MS | 105 | 0.80 | 0.34 | 0 | 1 | 210 | 0.77 | 0.35 | 0 | 1 |
| NSE | 64 | 0.72 | 0.38 | 0 | 1 | 266 | 0.74 | 0.37 | 0 | 1 |
| SS | 4 | 0.75 | 0.50 | 0 | 1 | 11 | 0.86 | 0.25 | 0 | 1 |
| All disciplines | 173 | 0.77 | 0.36 | 0 | 1 | 487 | 0.76 | 0.36 | 0 | 1 |

| Desk rejection rate of NEJM.C. | Women | | | | | Men | | | | |
|------------------------------|-----|------|-----------|-----|-----|------|------|-----------|-----|-----|
|  | Obs | Mean | Std. dev. | Min | Max | Obs | Mean | Std. dev. | Min | Max |
| MS | 161 | 0.68 | 0.40 | 0 | 1 | 320 | 0.67 | 0.40 | 0 | 1 |
| NSE | 9 | 0.72 | 0.44 | 0 | 1 | 28 | 0.71 | 0.43 | 0 | 1 |
| SS | 6 | 0.83 | 0.26 | 0.5 | 1 | 13 | 0.88 | 0.30 | 0 | 1 |
| All disciplines | 176 | 0.68 | 0.39 | 0 | 1 | 361 | 0.68 | 0.40 | 0 | 1 |

**Appendix 1—table 14.** Question: 'For each of the journals listed in the table below, please indicate the appropriate number of manuscripts rejected'—linear regression.

| Journal | Discipline | Coefficient | Std. dev. | t | p > \|t\| | 95% CI lower | 95% CI upper | obs |
|---------|-----------|-------------|-----------|------|-------|--------------|--------------|-----|
|  | MS | –0.03 | 0.02 | –1.33 | 0.185 | –0.07 | 0.01 | 809 |
|  | NSE | 0.03 | 0.03 | 1.01 | 0.314 | –0.02 | 0.07 | 928 |
|  | SS | 0.00 | 0.05 | –0.09 | 0.928 | –0.09 | 0.09 | 119 |
| S.N.P. | All disciplines | 0.00 | 0.02 | –0.31 | 0.757 | –0.04 | 0.03 | 1856 |
|  | MS | 0.04 | 0.04 | 0.96 | 0.336 | –0.04 | 0.12 | 311 |
|  | NSE | –0.03 | 0.05 | –0.61 | 0.542 | –0.14 | 0.07 | 326 |
|  | SS | –0.14 | 0.20 | –0.71 | 0.494 | –0.59 | 0.30 | 15 |
| NC.SA. | All disciplines | 0.01 | 0.03 | 0.17 | 0.867 | –0.06 | 0.07 | 652 |

*Appendix 1—table 14 Continued on next page*

*Appendix 1—table 14 Continued*

| Journal | Discipline | Coefficient | Std. dev. | t | p > |t| | 95% CI lower | 95% CI upper | obs |
|---|---|---|---|---|---|---|---|---|
| | MS | 0.02 | 0.04 | 0.37 | 0.708 | –0.06 | 0.09 | 467 |
| | NSE | 0.03 | 0.17 | 0.18 | 0.862 | –0.32 | 0.38 | 37 |
| | SS | –0.09 | 0.17 | –0.53 | 0.607 | –0.45 | 0.27 | 19 |
| NEJM.C. | All disciplines | 0.01 | 0.04 | 0.33 | 0.741 | –0.06 | 0.09 | 523 |

## Reasons for not considering submitting to top journals

**Appendix 1—table 15.** Question: 'Please indicate the reason(s) why you did not consider submitting a manuscript to journal name taken from response to the previous question'—number of respondents and percentage selecting response by discipline and gender.

| Reasons for not submitting to S.N.P. | MS | | | | NSE | | | | SS | | | |
|---|---|---|---|---|---|---|---|---|---|---|---|---|
| | Women | | Men | | Women | | Women | | Men | | Women | |
| | Total | %Yes | Total | %Yes | Total | %Yes | Total | %Yes | Total | %Yes | Total | %Yes |
| Work was not of high enough quality | 501 | 36% | 553 | 41% | 328 | 37% | 998 | 27% | 109 | 33% | 110 | 30% |
| Work fell outside the scope of the journal | 501 | 61% | 553 | 55% | 328 | 46% | 998 | 49% | 109 | 72% | 110 | 59% |
| Work was not groundbreaking or sufficiently novel | 501 | 52% | 553 | 50% | 328 | 53% | 998 | 44% | 109 | 53% | 110 | 40% |
| Work would fit better in a more specialized journal | 501 | 56% | 553 | 63% | 328 | 64% | 998 | 70% | 109 | 58% | 110 | 69% |
| Work would reach a wider audience in another journal | 501 | 9% | 553 | 13% | 328 | 16% | 998 | 17% | 109 | 7% | 110 | 8% |
| Co-authors wished to submit the manuscript elsewhere | 501 | 7% | 553 | 7% | 328 | 9% | 998 | 5% | 109 | 6% | 110 | 5% |
| I was advised against submitting to this journal | 501 | 6% | 553 | 4% | 328 | 8% | 998 | 5% | 109 | 6% | 110 | 5% |
| Reasons for not submitting to NC.SA. | MS | | | | NSE | | | | SS | | | |
| | Women | | Men | | Women | | Women | | Men | | Women | |
| | n | %Yes | n | %Yes | n | %Yes | n | %Yes | n | %Yes | n | %Yes |
| Work was not of high enough quality | 667 | 22% | 894 | 23% | 427 | 21% | 1467 | 15% | 157 | 12% | 180 | 17% |
| Work fell outside the scope of the journal | 667 | 46% | 894 | 37% | 427 | 33% | 1467 | 34% | 157 | 61% | 180 | 53% |
| Work was not groundbreaking or sufficiently novel | 667 | 32% | 894 | 29% | 427 | 37% | 1467 | 28% | 157 | 29% | 180 | 23% |

*Appendix 1—table 15 Continued on next page*

*Appendix 1—table 15 Continued*

| Reasons for not submitting to S.N.P. | MS | | | | NSE | | | | SS | | | |
| | Women | | Men | | Women | | Women | | Men | | Women | |
| | Total | %Yes | Total | %Yes | Total | %Yes | Total | %Yes | Total | %Yes | Total | %Yes |
|---|---|---|---|---|---|---|---|---|---|---|---|---|
| Work would fit better in a more specialized journal | 667 | 42% | 894 | 47% | 427 | 53% | 1467 | 52% | 157 | 39% | 180 | 51% |
| Work would reach a wider audience in another journal | 667 | 9% | 894 | 14% | 427 | 13% | 1467 | 17% | 157 | 10% | 180 | 11% |
| Co-authors wished to submit the manuscript elsewhere | 667 | 9% | 894 | 9% | 427 | 10% | 1467 | 8% | 157 | 6% | 180 | 4% |
| I was advised against submitting to this journal | 667 | 4% | 894 | 4% | 427 | 3% | 1467 | 3% | 157 | 3% | 180 | 2% |

| Reasons for not submitting to NEJM.C. | MS | | | | NSE | | | | SS | | | |
| | Women | | Men | | Women | | Women | | Men | | Women | |
| | n | %Yes | n | %Yes | n | %Yes | n | %Yes | n | %Yes | n | %Yes |
|---|---|---|---|---|---|---|---|---|---|---|---|---|
| Work was not of high enough quality | 602 | 34% | 791 | 31% | 503 | 6% | 1801 | 4% | 151 | 15% | 180 | 8% |
| Work fell outside the scope of the journal | 602 | 83% | 791 | 80% | 503 | 84% | 1801 | 86% | 151 | 93% | 180 | 90% |
| Work was not groundbreaking or sufficiently novel | 602 | 38% | 791 | 33% | 503 | 8% | 1801 | 5% | 151 | 21% | 180 | 12% |
| Work would fit better in a more specialized journal | 602 | 37% | 791 | 35% | 503 | 13% | 1801 | 13% | 151 | 24% | 180 | 27% |
| Work would reach a wider audience in another journal | 602 | 5% | 791 | 7% | 503 | 3% | 1801 | 4% | 151 | 5% | 180 | 6% |
| Co-authors wished to submit the manuscript elsewhere | 602 | 5% | 791 | 6% | 503 | 2% | 1801 | 2% | 151 | 5% | 180 | 3% |
| I was advised against submitting to this journal | 602 | 4% | 791 | 4% | 503 | 0% | 1801 | 0% | 151 | 2% | 180 | 1% |

**Appendix 1—table 16.** Question: 'Please indicate the reason(s) why you did not consider submitting a manuscript to journal name taken from response to the previous question'—odds ratio (women to men) values.

| Reasons why did not consider submitting papers to top journals | S.N.P. | | | | NC.SA. | | | | NEJM.C. | | | |
|---|---|---|---|---|---|---|---|---|---|---|---|---|
| | MS | NSE | SS | All displ. | MS | NSE | SS | All displ. | MS | NSE | SS | All displ. |
| Work was not of high enough quality | 0.83 | 1.59† | 1.04 | 1.12 | 0.91 | 1.51† | 0.60 | 1.08 | 1.10 | 1.51 | 1.69 | 1.21 |
| Work fell outside the scope of the journal | 1.27 | 0.94 | 1.84* | 1.14 | 1.42† | 1.01 | 1.41 | 1.24† | 1.20 | 0.92 | 1.41 | 1.08 |
| Work was not groundbreaking or sufficiently novel | 1.24* | 1.47† | 1.69* | 1.26† | 1.13 | 1.41† | 1.27 | 1.26† | 1.18 | 1.60* | 1.71 | 1.31† |
| Work would fit better in a more specialized journal | 0.74* | 0.72* | 0.58 | 0.72* | 0.82 | 1.02 | 0.60* | 0.87 | 1.06 | 1.01 | 0.81 | 1.01 |
| Work would reach a wider audience in another journal | 0.61* | 0.93 | 0.92 | 0.78 | 0.58† | 0.74 | 0.90 | 0.68† | 0.71 | 0.81 | 0.80 | 0.76 |
| Co-authors wished to submit the manuscript elsewhere | 1.02 | 1.70* | 1.31 | 1.32 | 1.00 | 1.17 | 1.34 | 1.09 | 0.88 | 1.09 | 1.34 | 0.97 |
| I was advised against submitting to this journal | 1.30 | 1.69* | 1.37 | 1.48* | 0.87 | 0.77 | 1.73 | 0.87 | 1.24 | 1.11 | 1.52 | 1.25 |
| I was unaware of this journal | ‡No one reported they were unaware of the top journals | | | | | | | | | | | |

* indicates p < 0.05.

† indicates p < 001.

‡Insufficient sample.

## Part 3: Most cited
Journal submission behavior

**Appendix 1—table 17.** Question: 'For your published manuscript <Title of the respondent's most cited paper>, did you ever consider submitting it to a journal other than ?'—percentage by discipline, and gender.

| Discipline | Women | | | | Men | | | |
|---|---|---|---|---|---|---|---|---|
| | No | Yes | Total | %Yes | No | Yes | Total | %Yes |
| MS | 370 | 432 | 802 | 53.9% | 559 | 616 | 1175 | 52.4% |
| NSE | 273 | 248 | 521 | 47.6% | 1038 | 814 | 1852 | 44.0% |
| SS | 80 | 84 | 164 | 51.2% | 101 | 92 | 193 | 47.7% |
| All disciplines | 723 | 764 | 1487 | 51.4% | 1698 | 1522 | 3220 | 47.3% |

**Appendix 1—table 18.** Question: 'For your published manuscript <Title of the respondent's most cited paper>, did you ever consider submitting it to a journal other than ?'—odds ratio (women to men) values.

| Other journals | F/M odds ratio | Std. err | z | p > |z| | 95% CI lower | 95% CI upper | obs |
|---|---|---|---|---|---|---|---|
| MS | 1.07 | 0.100 | 0.67 | 0.50 | 0.89 | 1.28 | 1914 |
| NSE | 1.12 | 0.113 | 1.14 | 0.26 | 0.92 | 1.37 | 2351 |
| SS | 1.16 | 0.250 | 0.68 | 0.50 | 0.76 | 1.77 | 351 |
| All disciplines | 1.10 | 0.072 | 1.40 | 0.16 | 0.96 | 1.25 | 4616 |

**Appendix 1—table 19.** Question: 'Before publication of your manuscript,<Title of the respondent's most cited paper>, did you ever consider submitting it to the journals listed in the table below?'—percentage by discipline, journal, and gender.

| Consider submitting to S.N.P. | Women | | | | Men | | | |
|---|---|---|---|---|---|---|---|---|
| | No | Yes | Total | %Yes | No | Yes | Total | %Yes |
| MS | 721 | 76 | 797 | 9.5% | 1029 | 143 | 1172 | 12.2% |
| NSE | 478 | 40 | 518 | 7.7% | 1702 | 148 | 1850 | 8.0% |
| SS | 157 | 6 | 163 | 3.7% | 185 | 13 | 198 | 6.6% |
| All disciplines | 1356 | 122 | 1478 | 8.3% | 2916 | 304 | 3220 | 9.4% |

| Consider submitting to NC.SA. | Women | | | | Men | | | |
|---|---|---|---|---|---|---|---|---|
| | No | Yes | Total | %Yes | No | Yes | Total | %Yes |
| MS | 743 | 32 | 775 | 4.1% | 1082 | 46 | 1128 | 4.1% |
| NSE | 486 | 17 | 503 | 3.4% | 1747 | 47 | 1794 | 2.6% |
| SS | 161 | 1 | 162 | 0.6% | 194 | 0 | 194 | 0.0% |
| All disciplines | 1390 | 50 | 1440 | 3.5% | 3023 | 93 | 3116 | 3.0% |

| Consider submitting to NEJM.C. | Women | | | | Men | | | |
|---|---|---|---|---|---|---|---|---|
| | No | Yes | Total | %Yes | No | Yes | Total | %Yes |
| MS | 749 | 42 | 791 | 5.3% | 1078 | 73 | 1151 | 6.3% |
| NSE | 500 | 2 | 502 | 0.4% | 1788 | 3 | 1791 | 0.2% |
| SS | 163 | 0 | 163 | 0.0% | 192 | 2 | 194 | 1.0% |
| All disciplines | 1412 | 44 | 1456 | 3.0% | 3058 | 78 | 3136 | 2.5% |

**Appendix 1—table 20.** Question: 'Before publication of your manuscript,<Title of the respondent's most cited paper>, did you ever consider submitting it to the journals listed in the table below?'—odds ratio (women to men) values.

| Journal | | W/M odds ratio | Std. err | z | p > |z| | 95% CI lower | 95% CI upper | obs |
|---|---|---|---|---|---|---|---|---|
| | MS | 0.79 | 0.120 | −1.53 | 0.126 | 0.59 | 1.07 | 1907 |
| | NSE | 0.96 | 0.181 | −0.21 | 0.83 | 0.66 | 1.39 | 2342 |
| | SS | 0.54 | 0.275 | −1.21 | 0.227 | 0.20 | 1.47 | 355 |
| S.N.P. | All disciplines | 0.83 | 0.096 | −1.59 | 0.112 | 0.66 | 1.04 | 4604 |
| | MS | 1.01 | 0.240 | 0.05 | 0.961 | 0.64 | 1.61 | 1844 |
| | NSE | 1.05 | 0.321 | 0.16 | 0.873 | 0.58 | 1.91 | 2272 |
| | SS | Insufficient sample | | | | | | |
| NC.SA. | All disciplines | 1.05 | 0.196 | 0.25 | 0.801 | 0.73 | 1.51 | 4466 |

*Appendix 1—table 20 Continued*

| Journal | | W/M odds ratio | Std. err | z | p > |z| | 95% CI lower | 95% CI upper | obs |
|---|---|---|---|---|---|---|---|---|
| | MS | 0.84 | 0.170 | –0.87 | 0.386 | 0.56 | 1.25 | 1882 |
| | NSE | 2.44 | 2.246 | 0.97 | 0.333 | 0.40 | 14.84 | 2268 |
| | SS | Insufficient sample | | | | | | |
| NEJM.C. | All disciplines | 0.85 | 0.166 | –0.85 | 0.393 | 0.57 | 1.24 | 4501 |

**Appendix 1—table 21.** Question: 'Before publication of your manuscript,<Title of the respondent's most cited paper>, did you ever submit it to the journals listed in the table below?'—percentage by discipline, journal, and gender.

| | Women | | | | Men | | | |
|---|---|---|---|---|---|---|---|---|
| **Submitted to S.N.P.** | **No** | **Yes** | **Total** | **%Yes** | **No** | **Yes** | **Total** | **%Yes** |
| MS | 37 | 38 | 75 | 50.7% | 58 | 81 | 139 | 58.3% |
| NSE | 22 | 17 | 39 | 43.6% | 64 | 80 | 144 | 55.6% |
| SS | 1 | 4 | 5 | 80.0% | 5 | 8 | 13 | 61.5% |
| All disciplines | 60 | 59 | 119 | 49.6% | 127 | 169 | 296 | 57.1% |
| | Women | | | | Men | | | |
| Submitted to NC.SA. | No | Yes | Total | %Yes | No | Yes | Total | %Yes |
| MS | 19 | 14 | 33 | 42.4% | 25 | 27 | 52 | 51.9% |
| NSE | 13 | 7 | 20 | 35.0% | 35 | 21 | 56 | 37.5% |
| SS | 1 | 0 | 1 | 0.0% | 0 | 0 | 0 | 0 |
| All disciplines | 33 | 21 | 54 | 38.9% | 60 | 48 | 108 | 44.4% |
| | Women | | | | Men | | | |
| Submitted to NEJM.C. | No | Yes | Total | %Yes | No | Yes | Total | %Yes |
| MS | 29 | 13 | 42 | 31.0% | 40 | 31 | 71 | 43.7% |
| NSE | 2 | 0 | 2 | 0.0% | 2 | 0 | 2 | 0.0% |
| SS | 0 | 0 | 0 | 0 | 2 | 0 | 2 | 0.0% |
| All disciplines | 31 | 13 | 44 | 29.5% | 44 | 31 | 75 | 41.3% |

**Appendix 1—table 22.** Question: 'Before publication of your manuscript, <Title of the respondent's most cited paper>, did you ever submit it to the journals listed in the table below?'—odds ratio (women to men) values for the probability of submitting.

| Journal | Discipline | W/M odds ratio | Std. err | z | p > |z| | 95% CI lower | 95% CI upper | obs |
|---|---|---|---|---|---|---|---|---|
| | MS | 0.75 | 0.22 | –0.99 | 0.321 | 0.43 | 1.32 | 210 |
| | NSE | 0.65 | 0.24 | –1.19 | 0.234 | 0.31 | 1.33 | 180 |
| | SS | 2.50 | 3.14 | 0.73 | 0.465 | 0.21 | 29.25 | 18 |
| S.N.P. | All disciplines | 0.74 | 0.17 | –1.35 | 0.177 | 0.48 | 1.15 | 408 |
| | MS | 0.69 | 0.32 | –0.82 | 0.414 | 0.28 | 1.70 | 84 |
| | NSE | 1.06 | 0.61 | 0.10 | 0.918 | 0.35 | 3.25 | 73 |
| | SS | No one submitted to NC.SA. | | | | | | |
| NC.SA. | All disciplines | 0.80 | 0.29 | –0.61 | 0.540 | 0.40 | 1.62 | 157 |

*Appendix 1—table 22 Continued on next page*

*Appendix 1—table 22 Continued*

| Journal | Discipline | W/M odds ratio | Std. err | z | p > \|z\| | 95% CI lower | 95% CI upper | obs |
|---|---|---|---|---|---|---|---|---|
|  | MS | 0.55 | 0.23 | −1.42 | 0.157 | 0.24 | 1.26 | 111 |
|  | NSE | No one submitted to NEJM.C. |  |  |  |  |  |  |
|  | SS | No one submitted to NEJM.C. |  |  |  |  |  |  |
| NEJM.C. | All disciplines | 0.57 | 0.24 | −1.35 | 0.178 | 0.25 | 1.29 | 117 |

## Desk rejections

**Appendix 1—table 23.** Question: 'Was your manuscript <Title of the respondent's most cited paper> sent out for peer review before being rejected by the journals listed in the table below?'—percentage by discipline, journal, and gender.

| Sent out for peer review S.N.P. | Women | | | | Men | | | |
|---|---|---|---|---|---|---|---|---|
|  | No | Yes | Total | %DeskRej | No | Yes | Total | %DeskRej |
| MS | 26 | 10 | 36 | 72.2% | 62 | 19 | 81 | 76.5% |
| NSE | 14 | 3 | 17 | 82.4% | 58 | 22 | 80 | 72.5% |
| SS | 3 | 1 | 4 | 75.0% | 8 | 0 | 8 | 100.0% |
| All disciplines | 43 | 14 | 57 | 75.4% | 128 | 41 | 169 | 75.7% |

| Sent out for peer review NC.SA. | Women | | | | Men | | | |
|---|---|---|---|---|---|---|---|---|
|  | No | Yes | Total | %DeskRej | No | Yes | Total | %DeskRej |
| MS | 12 | 2 | 14 | 85.7% | 21 | 7 | 28 | 75.0% |
| NSE | 7 | 1 | 8 | 87.5% | 17 | 4 | 21 | 81.0% |
| SS | No responses |  |  |  |  |  |  |  |
| All disciplines | 19 | 3 | 22 | 86.4% | 31 | 11 | 49 | 77.6% |

| Sent out for peer review NEJM.C. | Women | | | | Men | | | |
|---|---|---|---|---|---|---|---|---|
|  | No | Yes | Total | %DeskRej | No | Yes | Total | %DeskRej |
| MS | 5 | 8 | 13 | 38.5% | 27 | 5 | 32 | 84.4% |
| NSE |  |  |  |  |  |  |  |  |
| SS | No responses |  |  |  |  |  |  |  |

**Appendix 1—table 24.** Question: 'Was your manuscript <Title of the respondent's most cited paper> sent out for peer review before being rejected by the journals listed in the table below?'—odds ratio (women to men) values.

| Journal | Discipline | W/M odds ratio | Std. err | z | p > \|z\| | 95% CI lower | 95% CI upper | obs |
|---|---|---|---|---|---|---|---|---|
|  | MS | 0.48 | 0.66 | 0.74 | 0.46 | 0.56 | 3.52 | 114 |
|  | NSE | 1.41 | 0.33 | −1.05 | 0.29 | 0.12 | 1.88 | 95 |
|  | SS | Insufficient sample |  |  |  |  |  |  |
| S.N.P. | All disciplines | 1.13 | 0.41 | 0.32 | 0.75 | 0.55 | 2.31 | 221 |
|  | MS | 0.43 | 0.38 | −0.95 | 0.34 | 0.07 | 2.46 | 36 |
|  | NSE | 0.70 | 0.90 | −0.28 | 0.78 | 0.06 | 8.82 | 27 |
|  | SS | No responses |  |  |  |  |  |  |
| NC.SA. | All disciplines | 0.50 | 0.37 | −0.95 | 0.35 | 0.12 | 2.10 | 63 |

*Appendix 1—table 24 Continued on next page*

*Appendix 1—table 24 Continued*

| Journal | Discipline | W/M odds ratio | Std. err | z | p > \|z\| | 95% CI lower | 95% CI upper | obs |
|---|---|---|---|---|---|---|---|---|
|  | MS | 13.07 | 10.96 | 3.07 | 0.00 | 2.53 | 67.60 | 44 |
|  | NSE |  |  |  |  |  |  |  |
| NEJM.C. | SS | No responses |  |  |  |  |  |  |

## Reasons for not considering submitting to top journals

**Appendix 1—table 25.** Question: 'Please indicate the reason(s) why you did not consider submitting your manuscript <Title of the respondent's most cited paper> to the journals listed below.'—number of respondents and percentage selecting response by discipline and gender.

| Reasons for not submitting to S.N.P. | MS Women | | MS Men | | NSE Women | | NSE Men | | SS Women | | SS Men | |
|---|---|---|---|---|---|---|---|---|---|---|---|---|
|  | n | %Yes | n | %Yes | n | %Yes | n | %Yes | n | %Yes | n | %Yes |
| Work was not of high enough quality | 719 | 25% | 1027 | 30% | 475 | 21% | 1696 | 18% | 156 | 27% | 185 | 18% |
| Work fell outside the scope of the journal | 719 | 45% | 1027 | 44% | 475 | 35% | 1696 | 40% | 156 | 62% | 185 | 60% |
| Work was not groundbreaking or sufficiently novel | 719 | 45% | 1027 | 44% | 475 | 48% | 1696 | 39% | 156 | 39% | 185 | 30% |
| Work would fit better in a more specialized journal | 719 | 53% | 1027 | 53% | 475 | 60% | 1696 | 63% | 156 | 46% | 185 | 57% |
| Work would reach a wider audience in another journal | 719 | 8% | 1027 | 9% | 475 | 12% | 1697 | 14% | 156 | 8% | 185 | 11% |
| Co-authors wished to submit the manuscript elsewhere | 719 | 7% | 1027 | 6% | 475 | 8% | 1697 | 4% | 156 | 6% | 185 | 3% |
| I was advised against submitting to this journal | 719 | 5% | 1027 | 3% | 475 | 3% | 1696 | 2% | 156 | 2% | 185 | 2% |

| Reasons for not submitting to NC.SA. | MS Women | | MS Men | | NSE Women | | NSE Women | | SS Men | | SS Women | |
|---|---|---|---|---|---|---|---|---|---|---|---|---|
|  | n | %Yes | n | %Yes | n | %Yes | n | %Yes | n | %Yes | n | %Yes |
| Work was not of high enough quality | 732 | 19% | 1076 | 22% | 481 | 14% | 1731 | 13% | 159 | 14% | 189 | 12% |
| Work fell outside the scope of the journal | 732 | 41% | 1076 | 38% | 481 | 31% | 1731 | 33% | 159 | 63% | 189 | 58% |
| Work was not groundbreaking or sufficiently novel | 732 | 34% | 1076 | 32% | 481 | 38% | 1731 | 30% | 159 | 29% | 189 | 21% |

*Appendix 1—table 25 Continued on next page*

*Appendix 1—table 25 Continued*

| Reasons for not submitting to S.N.P. | MS | | | | NSE | | | | SS | | | |
|---|---|---|---|---|---|---|---|---|---|---|---|---|
| | Women | | Men | | Women | | Men | | Women | | Men | |
| | n | %Yes | n | %Yes | n | %Yes | n | %Yes | n | %Yes | n | %Yes |
| Work would fit better in a more specialized journal | 732 | 46% | 1076 | 45% | 481 | 52% | 1731 | 55% | 159 | 32% | 189 | 43% |
| Work would reach a wider audience in another journal | 732 | 8% | 1076 | 11% | 481 | 10% | 1731 | 12% | 159 | 5% | 189 | 7% |
| Co-authors wished to submit the manuscript elsewhere | 732 | 6% | 1076 | 6% | 481 | 6% | 1732 | 5% | 159 | 8% | 189 | 2% |
| I was advised against submitting to this journal | 732 | 2% | 1076 | 2% | 481 | 2% | 1731 | 2% | 159 | 1% | 189 | 1% |

| Reasons for not submitting to NEJM.C. | MS | | | | NSE | | | | SS | | | |
|---|---|---|---|---|---|---|---|---|---|---|---|---|
| | Women | | Men | | Women | | Women | | Men | | Women | |
| | n | %Yes | n | %Yes | n | %Yes | n | %Yes | n | %Yes | n | %Yes |
| Work was not of high enough quality | 738 | 25% | 1073 | 28% | 495 | 5% | 1771 | 5% | 161 | 15% | 187 | 7% |
| Work fell outside the scope of the journal | 738 | 72% | 1073 | 75% | 495 | 85% | 1772 | 86% | 161 | 91% | 187 | 86% |
| Work was not groundbreaking or sufficiently novel | 738 | 37% | 1073 | 36% | 495 | 9% | 1771 | 7% | 161 | 18% | 187 | 11% |
| Work would fit better in a more specialized journal | 738 | 44% | 1073 | 42% | 495 | 16% | 1771 | 18% | 161 | 25% | 187 | 30% |
| Work would reach a wider audience in another journal | 738 | 6% | 1073 | 6% | 495 | 4% | 1771 | 4% | 161 | 4% | 187 | 7% |
| Co-authors wished to submit the manuscript elsewhere | 738 | 5% | 1073 | 4% | 495 | 2% | 1771 | 1% | 161 | 5% | 187 | 2% |
| I was advised against submitting to this journal | 738 | 2% | 1073 | 2% | 495 | 0% | 1771 | 0% | 161 | 1% | 187 | 0% |

**Appendix 1—table 26.** Question: 'Please indicate the reason(s) why you did not consider submitting your manuscript <Title of the respondent's most cited paper> to the journals listed below.'—odds ratio (women to men) values.

| Reasons why did not consider submitting the most cited papers to top journals | S.N.P. | | | | NC.SA. | | | | NEJM.C. | | | |
|---|---|---|---|---|---|---|---|---|---|---|---|---|
| | MS | NSE | SS | All | MS | NSE | SS | All | MS | NSE | SS | All |
| Work was not of high enough quality | 0.75* | 1.17 | 1.63 | 0.96 | 0.86 | 1.00 | 1.31 | 0.93 | 0.83 | 0.94 | 2.36* | 0.91 |
| Work fell outside the scope of the journal | 1.05 | 0.82 | 1.15 | 0.95 | 1.09 | 0.93 | 1.30 | 1.04 | 0.86 | 0.94 | 1.52 | 0.92 |
| Work was not groundbreaking or sufficiently novel | 1.02 | 1.39 ‡ | 1.48 | 1.19* | 1.05 | 1.28* | 1.81 | 1.14 | 1.05 | 1.28 | 1.81 | 1.14 |
| Work would fit better in a more specialized journal | 0.99 | 0.88 | 0.66 | 0.90 | 1.00 | 0.90 | 0.64 | 0.92 | 1.08 | 0.92 | 0.79 | 1.00 |
| Work would reach a wider audience in another journal | 0.96 | 0.87 | 0.71 | 0.89 | 0.72 | 0.87 | 0.76 | 0.79 | 0.98 | 0.85 | 0.55 | 0.88 |
| Co-authors wished to submit the manuscript elsewhere | 1.13 | 1.57* | 2.26 | 1.35* | 0.99 | 1.06 | 4.53* | 1.12 | 1.14 | 1.41 | 2.07 | 1.26 |
| I was advised against submitting to this journal | 1.75* | 1.51 | 0.97 | 1.59* | 0.96 | 1.06 | 2.27 | 1.03 | 1.47 | 3.52 | † | 1.67 |
| I was unaware of this journal | †No one reported they were unaware of the top journals | | | | | | | | | | | |

* indicates $p < 0.05$.
†Insufficient sample.
‡ indicates $p < 0.01$.

# Part 4: Perception of the quality of research

**Appendix 1—table 27.** Question: 'Compared to my peers, I feel that the quality of my research is'—average rank and count by discipline and gender.

**Women**

| Discipline | Average quality rank | Std. dev | #Good and Excellent | #Average | #Fair and Poor | Total |
|---|---|---|---|---|---|---|
| MS | 4.2 | 0.69 | 574 | 74 | 8 | 656 |
| NSE | 4.0 | 0.67 | 352 | 60 | 8 | 420 |
| SS | 4.1 | 0.62 | 135 | 18 | 0 | 153 |
| All disciplines | 4.1 | 0.68 | 1061 | 152 | 16 | 1229 |

**Men**

| Discipline | Average quality rank | Std. dev | #Good and Excellent | #Average | #Fair and Poor | Total |
|---|---|---|---|---|---|---|
| MS | 4.1 | 0.7 | 841 | 126 | 7 | 974 |
| NSE | 4.2 | 0.6 | 1363 | 185 | 9 | 1557 |

*Appendix 1—table 27 Continued on next page*

*Appendix 1—table 27 Continued*

**Women**

| | | | | | | |
|---|---|---|---|---|---|---|
| SS | 4.1 | 0.8 | 151 | 25 | 6 | 182 |
| All disciplines | 4.1 | 0.7 | 2355 | 336 | 22 | 2713 |

Note: Excellent (5)–Poor (1).

**Appendix 1—table 28.** Question: 'Compared to my peers, I feel that the quality of my research is'—ordinal logistic (women to men), controlled for rank.

| Disciplines | Odds ratio | Std. err | z | p > |z| | 95% CI lower | 95% CI upper | obs |
|---|---|---|---|---|---|---|---|
| MS | 1.18 | 0.13 | 2.40 | 0.07 | 0.99 | 1.40 | 1630 |
| NSE | 0.83 | 0.09 | −1.64 | 0.04 | 0.67 | 0.99 | 1977 |
| SS | 1.13 | 0.24 | 0.56 | 0.58 | 0.74 | 1.71 | 335 |
| All disciplines | 1.06 | 0.07 | 0.87 | 0.38 | 0.93 | 1.22 | 3,942 |

