## [Editor Report · eLife Assessment]

This **convincing** study, which is based on a survey of researchers, finds that women are less likely than men to submit articles to elite journals. It also finds that there is no relation between gender and reported desk rejection. The study is an **important** contribution to work on gender bias in the scientific literature.

---

## [Referee Report · Joint Public Review]

Summary from an earlier round of review:

This paper summarises responses from a survey completed by around 5,000 academics on their manuscript submission behaviours. The authors find several interesting stylised facts, including (but not limited to):- Women are less likely to submit their papers to highly influential journals (e.g., Nature, Science and PNAS).

- Women are more likely to cite the demands of co-authors as a reason why they didn’t submit to highly influential journals.

- Women are also more likely to say that they were advised not to submit to highly influential journals.

The paper highlights an important point, namely that the submission behaviours of men and women scientists may not be the same (either due to preferences that vary by gender, selection effects that arise earlier in scientists’ careers or social factors that affect men and women differently and also influence submission patterns). As a result, simply observing gender differences in acceptance rates - or a lack thereof - should not be automatically interpreted as as evidence for or against discrimination (broadly defined) in the peer review process.

Editor’s note: This is the third version of this article.

Comments made during the peer review of the second version, along with author’s responses to these comments, are available below. Revisions made in response to these comments include changing the colour scheme used for the figures to make the figures more accessible for readers with certain forms of colour blindness.

Comments made during the peer review of the first version, along with author’s responses to these comments, are available with previous versions of the article.

---

## [Author Response]

The following is the authors’ response to the previous reviews

**Reviewer #1 (Public review):**
SummaryThis paper summarises responses from a survey completed by around 5,000 academics on their manuscript submission behaviours. The authors find several interesting stylised facts, including (but not limited to):Women are less likely to submit their papers to highly influential journals (e.g., Nature, Science and PNAS).Women are more likely to cite the demands of co-authors as a reason why they didn't submit to highly influential journals.Women are also more likely to say that they were advised not to submit to highly influential journals.The paper highlights an important point, namely that the submission behaviours of men and women scientists may not be the same (either due to preferences that vary by gender, selection effects that arise earlier in scientists' careers or social factors that affect men and women differently and also influence submission patterns). As a result, simply observing gender differences in acceptance rates - or a lack thereof - should not be automatically interpreted as as evidence for or against discrimination (broadly defined) in the peer review process.Major commentsWhat do you mean by bias?In the second paragraph of the introduction, it is claimed that "if no biases were present in the case of peer review, then we should expect the rate with which members of less powerful social groups enjoy successful peer review outcomes to be proportionate to their representation in submission rates." There are a couple of issues with this statement.First, the authors are implicitly making a normative assumption that manuscript submission and acceptance rates *should* be equalised across groups. This may very well be the case, but there can also be valid reasons - even when women are not intrinsically better at research than men - why a greater fraction of female-authored submissions are accepted relative to male-authored submissions (or vice versa). For example, if men are more likely to submit their less ground-breaking work, then one might reasonably expect that they experience higher rejection rates compared to women, conditional on submission.

We do assume that normative statement: unless we believe that men’s papers are intrinsically better than women’s papers, the acceptance rate should be the same. But the referee is right: we have no way of controlling for the intrinsic quality of the work of men and women. That said, our manuscript does not show that there is a different acceptance rate for men and women; it shows that women are less likely to submit papers to a subset of journals that are of a lower Journal Impact Factor, controlling for their most cited paper, in an attempt to control for intrinsic quality of the manuscripts.

Second, I assume by "bias", the authors are taking a broad definition, i.e., they are not only including factors that specifically relate to gender but also factors that are themselves independent of gender but nevertheless disproportionately are associated with one gender or another (e.g., perhaps women are more likely to write on certain topics and those topics are rated more poorly by (more prevalent) male referees; alternatively, referees may be more likely to accept articles by authors they've met before, most referees are men and men are more likely to have met a given author if he's male instead of female). If that is the case, I would define more clearly what you mean by bias. (And if that isn't the case, then I would encourage the authors to consider a broader definition of "bias"!)

Yes, the referee is right that we are taking a broad definition of bias. We provide a definition of bias on page 3, line 92. This definition is focused on differential evaluation which leads to differential outcomes. We also hedge our conversation (e.g., page 3, line 104) to acknowledge that observations of disparities may only be an indicator of potential bias, as many other things could explain the disparity. In short, disparities are a necessary but insufficient indicator of bias. We add a line in the introduction to reinforce this. The only other reference to the term bias comes on page 10, line 276. We add a reference to Lee here to contextualize.

Identifying policy interventions is not a major contribution of this paperI would take out the final sentence in the abstract. In my opinion, your survey evidence isn't really strong enough to support definitive policy interventions to address the issue and, indeed, providing policy advice is not a major - or even minor - contribution of your paper. (Basically, I would hope that someone interested in policy interventions would consult another paper that much more thoughtfully and comprehensively discusses the costs and benefits of various interventions!) While it's fine to briefly discuss them at the end of your paper - as you currently do - I wouldn't highlight that in the abstract as being an important contribution of your paper.

We thank the referee for this comment. While we agree that our results do not lead to definitive policy interventions, we believe that our findings point to a phenomenon that should be addressed through policy interventions. Given that some interventions are proposed in our conclusion, we feel like stating this in the abstract is coherent.

Minor commentsWhat is the rationale for conditioning on academic rank and does this have explanatory power on its own - i.e., does it at least superficially potentially explain part of the gender gap in intention to submit?

Thank you for this thoughtful question. We conditioned on academic rank in all regression analyses to account for structural differences in career stage that may potentially influence submission behaviors. Academic rank (e.g., assistant, associate, full professor) is a key determinant of publishing capacity and strategic considerations, such as perceived likelihood of success at elite journals, tolerance for risk, and institutional expectations for publication venues.

Importantly, academic rank is also correlated with gender due to cumulative career disadvantages that contribute to underrepresentation of women at more senior levels. Failing to adjust for rank would conflate gender effects with differences attributable to career stage. By including rank as a covariate, we aim to isolate gender-associated patterns in submission behavior within comparable career stages, thereby producing a more precise estimate of the gender effect.

Regarding explanatory power, academic rank does indeed contribute significantly to model fit across our analyses, indicating that it captures meaningful variation in submission behavior. However, even after adjusting for rank, we continue to observe significant gender differences in submission patterns in several disciplines. This suggests that while academic rank explains part of the variation, it does not fully account for the gender gap—highlighting the importance of examining other structural and behavioral factors that shape the publication trajectory.

**Reviewer #2 (Public review):**
Basson et al. present compelling evidence supporting a gender disparity in article submission to "elite" journals. Most notably, they found that women were more likely to avoid submitting to one of these journals based on advice from a colleague/mentor. Overall, this work is an important addition to the study of gender disparities in the publishing process.I thank the authors for addressing my concerns.
**Reviewer #4 (Public review):**
Main strengthsThe topic of the MS is very relevant given that across the sciences/academia, genders are unevenly represented, which has a range of potential negative consequences. To change this, we need to have the evidence on what mechanisms cause this pattern. Given that promotion and merit in academia are still largely based on the number of publications and the impact factor, one part of the gap likely originates from differences in publication rates of women compared to men.Women are underrepresented compared to men in journals with a high impact factor. While previous work has detected this gap and identified some potential mechanisms, the current MS provides strong evidence that this gap might be due to a lower submission rate of women compared to men, rather than the rejection rates. These results are based on a survey of close to 5000 authors. The survey seems to be conducted well (though I am not an expert in surveys), and data analysis is appropriate to address the main research aims. It was impossible to check the original data because of the privacy concerns.Interestingly, the results show no gender bias in rejection rates (desk rejection or overall) in three high-impact journals (Science, Nature, PNAS). However, submission rates are lower for women compared to men, indicating that gender biases might act through this pathway. The survey also showed that women are more likely to rate their work as not groundbreaking and are advised not to submit to prestigious journals, indicating that both intrinsic and extrinsic factors shape women's submission behaviour.With these results, the MS has the potential to inform actions to reduce gender bias in publishing, but also to inform assessment reform at a larger scale.I do not find any major weaknesses in the revised manuscript.
**Reviewer #4 (Recommendations for the authors):**
(1) Colour schemes of the Figures are not adjusted for colour-blindness (red-green is a big NO), some suggestions can be found here https://www.nceas.ucsb.edu/sites/default/files/2022-06/Colorblind%20Safe%20Color%20Schemes.pdf

We appreciate the suggestion. We’ve adjusted the colors in the manuscript to be color-blind friendly using one of the colorblind safe palettes suggested by the reviewer.

(2) I do not think that the authors have fully addressed the comment about APCs and the decision to submit, given that PNAS has publication charges that amount to double of someone's monthly salary. I would add a sentence or two to explain that publication charges should not be a factor for Nature and Science, but might be for PNAS.

While APCs are definitely a factor affecting researchers’ submission behavior, it is mostly does so for lower prestige journals rather than for the three elite journals analyzed here. As mentioned in the previous round of revisions, Nature and Science have subscription options. And PNAS authors without funding have access to waivers: https://www.pnas.org/author-center/publication-charges

(3) Line 268, the first suggestion here is not something that would likely work. Thus, I would not put it as the first suggestion.

We made the suggested change.

(4) Data availability - remove AND in 'Aggregated and de-identified data' because it sounds like both are shared. Suggest writing: 'Aggregated, de-identified data..'. I still suggest sharing data/code in a trusted repository (e.g. Dryad, ZENODO...) rather than on GitHub, as per the current recommendation on the best practices for data sharing.

Thank you for your comment regarding data availability. Due to IRB restrictions and the conditions of our ethics approval, we are not permitted to share the survey data used in this study. However, to support transparency and reproducibility, we have made all analysis code available on Zenodo at https://doi.org/10.5281/zenodo.16327580. In addition, we have included a synthetic dataset with the same structure as the original survey data but containing randomly generated values. This allows others to understand the data structure and replicate our analysis pipeline without compromising participant confidentiality.